# PROPORTIONAL AMPLITUDE SPECTRUM TRAINING AUGMENTATION FOR SYN-TO-REAL DOMAIN GENERALIZATION

## ABSTRACT

Synthetic data offers the promise of cheap and bountiful training data for settings where lots of labeled real-world data for some tasks is unavailable. However, models trained on synthetic data significantly underperform on real-world data. In this paper, we propose Proportional Amplitude Spectrum Training Augmentation (PASTA), a simple and effective augmentation strategy to improve out-of-the-box synthetic-to-real (syn-to-real) generalization performance. PASTA involves perturbing the amplitude spectrums of the synthetic images in the Fourier domain to generate augmented views. We design PASTA to perturb the amplitude spectrums in a structured manner such that high-frequency components are perturbed relatively more than the low-frequency ones. For the tasks of semantic segmentation (GTAV→Real), object detection (Sim10K→Real), and object recognition (VisDA-C Syn→Real), across a total of 5 syn-to-real shifts, we find that PASTA either outperforms or is consistently competitive with more complex state-of-the-art methods while being complementary to other generalization approaches.

## 1 INTRODUCTION

Performant deep models for complex tasks heavily rely on access to substantial labeled data during training. However, gathering labeled real-world data can be expensive and often only captures a portion of the real-world seen at test time. Therefore, training models on synthetic data to better generalize to diverse real-world data has emerged as a popular alternative. However, models trained on synthetic data have a hard time generalizing to real world data – e.g., the performance of a vanilla DeepLabv3+ (Chen et al., 2018a) (ResNet-50 backbone) architecture on semantic segmentation drops from 73.45% mIoU on GTAV to 28.95% mIoU on Cityscapes for the same set of classes. Several approaches have been considered in prior work to tackle this problem.

In this paper, we propose an augmentation strategy, called Proportional Amplitude Spectrum Training Augmentation (PASTA), for the synthetic-to-real generalization problem. PASTA involves perturbing the amplitude spectrums of the source synthetic images in the Fourier domain. While prior work in domain generalization has considered augmenting images in the Fourier domain (Xu et al., 2021; Yang & Soatto, 2020; Huang et al., 2021a), they mostly rely on the observations that – (1) low-frequency bands of the amplitude spectrum tend to capture style information / low-level statistics (illumination, lighting, etc.) (Yang & Soatto, 2020) and (2) the corresponding phase spectrum tends to capture high-level semantic content (Oppenheim et al., 1979; Oppenheim & Lim, 1981; Piotrowski & Campbell, 1982; Hansen & Hess, 2007; Yang et al., 2020).

In addition to the observations from prior work, we make the observation that synthetic images have less diversity in the high-frequency bands of their amplitude spectrums compared to real images (see Sec. 3.2 for a detailed discussion). Motivated by these key observations, PASTA provides a structured way to perturb the amplitude spectrums of source synthetic images to ensure that a model is exposed to more variations in high-frequency components during training. We empirically observe that by relying on such a simple set of motivating observations, PASTA leads to significant improvements in synthetic-to-real generalization performance – e.g., out-of-the-box GTAV→Cityscapes generalization performance of a vanilla DeepLabv3+ (ResNet-50 backbone) semantic segmentation architecture improves from 28.95% mIoU to 44.12% mIoU.

PASTA involves the following steps. Given an input image, we apply 2D Fast Fourier Transform (FFT) to obtain the corresponding amplitude and phase spectrums in the Fourier domain. For every spatial frequency $(m, n)$ in the amplitude spectrum, we sample a multiplicative jitter value $\epsilon$ from

Figure 1: **PASTA augmentation samples.** Examples of images from different synthetic datasets when augmented using PASTA and RandAugment (Cubuk et al., 2020). Row 1 includes examples from GTAV and row 2 from VisDA-C.

$\mathcal{N}(1, \sigma^2[m, n])$ such that $\sigma[m, n]$ increases monotonically with $(m, n)$ (specifically $\sqrt{m^2 + n^2}$), thereby, ensuring that higher frequency components in the amplitude spectrum are perturbed more compared to the lower frequency components. The dependence of $\sigma[m, n]$ on $(m, n)$ can be controlled using a set of hyper-parameters that govern the degree of monotonicity. Finally, given the perturbed amplitude and the original phase spectrums, we can apply an inverse 2D Fast Fourier Transform (iFFT) to obtain the augmented image. Fig. 1 shows a few examples of augmentation by PASTA.

In terms of Fourier domain augmentations, closest to PASTA are perhaps the approaches – Amplitude Jitter (AJ) (Xu et al., 2021) and Amplitude Mixup (AM) (Xu et al., 2021). The overarching principle across these methods is to perturb only the amplitude spectrums of images (while keeping the phase spectrum unaffected) to ensure models are invariant to the applied perturbations. For instance, AM, which is a type of mixup strategy (Zhang et al., 2018; Verma et al., 2019), performs mixup between the amplitude spectrums of distinct intra-source images, while AJ uniformly perturbs the amplitude spectrums with a single jitter value $\epsilon$. Another frequency randomization technique, Frequency Space Domain Randomization (FSDR) (Huang et al., 2021a), first isolates domain variant and invariant frequency components by using SYNTHIA (Ros et al., 2016) (extra data) and ImageNet and then sets up a learning paradigm. Unlike these methods, PASTA applies fine-grained perturbations and does not involve sampling a separate mixup image or the use of any extra images. Instead, PASTA provides a simple strategy to perturb the amplitude spectrum of images in a structured way that leads to strong out-of-the-box generalization. We will release our code and data upon acceptance.

In summary, we make the following contributions.

- We introduce Proportional Amplitude Spectrum Training Augmentation (PASTA), a simple and effective augmentation strategy for synthetic-to-real generalization. PASTA involves perturbing the amplitude spectrums of synthetic images in the Fourier domain so as to expose a model to more variations in high-frequency components.
- We show that PASTA leads to considerable improvements or competitive results across three tasks – (1) Semantic Segmentation: GTAV → Cityscapes, Mapillary, BDD100k, (2) Object Detection: Sim10K → Cityscapes and (3) Object Recognition: VisDA-C Syn → Real – covering a total of 5 syn-to-real shifts across multiple backbones.
- We show that PASTA (1) often makes a baseline model competitive with prior state-of-the-art approaches relying on either specific architectural components, extra data, or objectives, (2) is complementary to said approaches and (3) is competitive with augmentation strategies like FACT (Xu et al., 2021) and RandAugment (Cubuk et al., 2020).

## 2 RELATED WORK

**Domain Generalization (DG).** DG involves training models on single or multiple labeled data sources to generalize well to novel test time data sources (unseen during training). Since its inception (Blanchard et al., 2011; Muandet et al., 2013), several approaches have been proposed to tackle the problem of domain generalization. These include – decomposing a model into domain invariant and specific components and utilizing the former to make predictions (Ghifary et al., 2015; Khosla et al., 2012), learning domain specific masks for generalization (Chattopadhyay et al., 2020), using meta-learning to train a robust model by mimicking the DG problem during training (Li et al., 2018; Wang et al., 2020; Balaji et al., 2018; Chen et al., 2022; Dou et al., 2019), manipulating feature statistics to augment training data (Zhou et al., 2021; Li et al., 2022; Nuriel et al., 2021), and using models crafted based on risk minimization formalisms (Arjovsky et al., 2019). Recently, properly tuned Empirical Risk Minimization (ERM) has proven to be a competitive DG approach (Gulrajani & Lopez-Paz, 2020) with follow-up work adopting various optimization and regularization techniques on top of ERM (Shi et al., 2021; Cha et al., 2021).

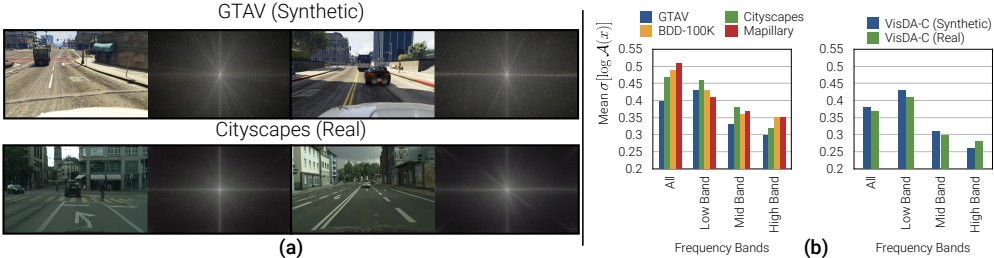

(a)

(b)

Figure 2: **Amplitude spectrum characteristics.** (a) Sample amplitude spectrums (lowest frequency at the center) for one channel of synthetic and real images. Note that the amplitude spectrums tend to follow a specific pattern – statistics of natural images have been found to exhibit the property that amplitude values tend to follow an inverse power law w.r.t. the frequency (Burton & Moorhead, 1987; Tolhurst et al., 1992), *i.e.*, roughly, the amplitude at frequency $f$, $\mathcal{A}(f) \propto \frac{1}{f^{\gamma}}$, for some $\gamma$ determined empirically. (b) Variations in amplitude values across images. Synthetic images have less variance in high-frequency components of the amplitude spectrum compared to real images.

**Single Domain Generalization (SDG).** Unlike DG which leverages diversity across domains for generalization, SDG considers generalizing from only one source domain. Some notable approaches for SDG involve using meta-learning (Qiao et al., 2020) by considering strongly augmented versions of source images as meta-target data (by exposing the model to increasingly distinct augmented views of the source data (Wang et al., 2021; Li et al., 2021)) and learning feature normalization schemes with auxiliary objectives (Fan et al., 2021).

**Synthetic-to-Real Generalization (syn-to-real).** Approaches specific to syn-to-real generalization in prior work (most relevant to our experimental settings) mostly consider either learning specific feature normalization schemes so that predictions are invariant to style characteristics (Pan et al., 2018; Choi et al., 2021), collecting external data to inject style information (Kim et al., 2021; Kundu et al., 2021), learning to optimize for robustness (Chen et al., 2020b), leveraging strong augmentations / domain randomization (Yue et al., 2019; Kundu et al., 2021) or using contrastive techniques to aid generalization (Chen et al., 2021). To adapt to real images from synthetic data (Chen et al., 2018b) trained Faster R-CNN in an adversarial manner, (Saito et al., 2019) leveraged adversarial alignment loss to emphasize globally similar images, (Chen et al., 2020a) proposed a method to harmonize transferability and discriminability of features in a hierarchical method, and (Vibashan et al., 2021) ensures category-aware feature alignment for learning domain-invariant features. We consider three of the most commonly studied settings for syn-to-real generalization in this paper – (1) Semantic Segmentation - GTAV→Real datasets, (2) Object Detection - Sim10K→Real and (3) Object Recognition - the VisDA-C (Peng et al., 2017) dataset. To appropriately characterize the "right" synthetic data for generalization, (Mishra et al., 2021) has recently considered tailoring synthetic data for downstream generalization.

**Fourier Domain Generalization, Adaptation and Robustness.** Prior work has considered augmenting images in the frequency domain as opposed to the pixel space. These approaches rely on the empirical observation (Oppenheim et al., 1979; Oppenheim & Lim, 1981; Piotrowski & Campbell, 1982; Hansen & Hess, 2007) that the phase component of the Fourier spectrum corresponds to the semantics of the image. PASTA is in line with this style of approach. Closest and perhaps most relevant to our work are that of (Xu et al., 2021; Yang & Soatto, 2020) which consider perturbing the amplitude spectrums of the source synthetic images. Building on top of (Xu et al., 2021), (Yang et al., 2021) adds a significance mask during linear interpolation of amplitudes. (Huang et al., 2021b) decomposes images into multiple frequency components and only perturbs components that capture little semantic information. (Wang et al., 2022) uses an encoder-decoder architecture to obtain high and low frequency features, and augments the image by adding random noise to the phase of high frequency features and to the amplitude of low frequency features. More generally, (Yin et al., 2019) finds that perturbations in the higher frequency domain increase robustness of models to high-frequency image corruptions.

## 3 METHOD

In this section, we first cover preliminaries and then describe our proposed approach.

### 3.1 PRELIMINARIES

**Problem Setup.** In this work, we study the single domain generalization (SDG) problem in the synthetic-to-real context. SDG typically involves training a model on a single labeled data source

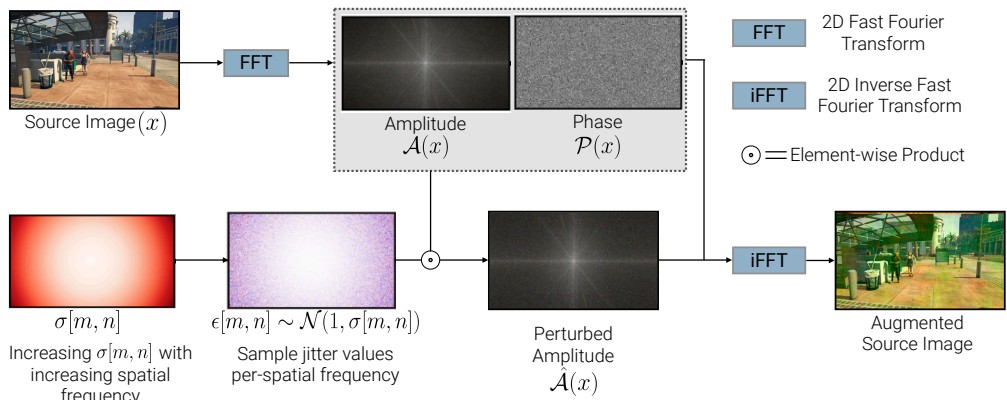

Figure 3: **PASTA.** The figure outlines the augmentation pipeline involved in PASTA. Given an image, we first apply a 2D Fast Fourier Transform (FFT) to obtain the amplitude and phase spectrums. Following this, the Amplitude spectrum is perturbed as outlined in Eqns. 4 and 5. Finally, we use the perturbed amplitude spectrum and the pristine phase spectrum to recover the augmented image by applying a 2D inverse FFT.

to generalize well to a target distribution(s), without access to target data during training. For our experiments, source data is synthetic and target data is real.

**Fourier Transform.** Consider a single-channel image $x \in \mathbb{R}^{H \times W}$. The Fourier transform for $x$ can be expressed as,

$$\mathcal{F}(x)[m, n] = \sum_{h=0}^{H-1} \sum_{w=0}^{W-1} x[h, w] \exp\left(-2\pi i \left(\frac{h}{H}m + \frac{w}{W}n\right)\right) \qquad (1)$$

where $i^2 = -1$. The inverse Fourier transform, $\mathcal{F}^{-1}(\cdot)$, that maps signals from the frequency domain to the image domain can be defined accordingly. Note that the Fourier spectrum $\mathcal{F}(x) \in \mathbb{C}^{H \times W}$. If $\mathbf{Re}(\mathcal{F}(x)[\cdot, \cdot])$ and $\mathbf{Im}(\mathcal{F}(x)[\cdot, \cdot])$ denote the real and imaginary parts of the Fourier spectrum, the corresponding amplitude ($\mathcal{A}(x)[\cdot]$) and phase ($\mathcal{P}(x)[\cdot]$) spectrums can be expressed as,

$$\mathcal{A}(x)[m, n] = \sqrt{\mathbf{Re}(\mathcal{F}(x)[m, n])^2 + \mathbf{Im}(\mathcal{F}(x)[m, n])^2} \qquad (2)$$

$$\mathcal{P}(x)[m, n] = \mathtt{arctan}\left(\frac{\mathbf{Im}(\mathcal{F}(x)[m, n])}{\mathbf{Re}(\mathcal{F}(x)[m, n])}\right) \qquad (3)$$

Without loss of generality, we will assume for the rest of this section that the amplitude and phase spectrums are zero-centered, *i.e.*, the low-frequency components (low $m, n$) have been shifted to the center (lowest frequency component is at the center). The Fourier transform and its inverse can be calculated efficiently using the Fast Fourier Transform (FFT) (Nussbaumer, 1981) algorithm. For an RGB image, we can obtain the Fourier spectrum (and $\mathcal{A}(x)[\cdot]$ and $\mathcal{P}(x)[\cdot]$) independently for each channel. For the following section, we will use a single channel image as a running example to describe our proposed approach.

## 3.2 PROPORTIONAL AMPLITUDE SPECTRUM TRAINING AUGMENTATION (PASTA)

We propose to create a set of augmented images through perturbations to the Fourier Amplitude spectrum. Following prior work (Oppenheim et al., 1979; Oppenheim & Lim, 1981; Piotrowski & Campbell, 1982; Hansen & Hess, 2007), our augmentations perturb only the amplitude spectrum and leave the phase untouched to roughly preserve the semantics of the scene. The key steps involved in generating an augmented image from an original image $x \in \mathbb{R}^{H \times W}$ are:

1. Translating $x$ to the Fourier domain to obtain the Fourier, amplitude and phase spectrums (see Eqns 1, 2, 3) – $\mathcal{F}(x), \mathcal{A}(x), \mathcal{P}(x)$

2. Perturbing "only" the amplitude spectrum via a perturbation function $g_\Lambda(\cdot)$ to obtain the perturbed amplitude spectrum $\hat{\mathcal{A}}(x) = g_\Lambda(\mathcal{A}(x))$

3. Applying an inverse Fourier transform, $\mathcal{F}^{-1}(\cdot)$, to the perturbed amplitude $\hat{\mathcal{A}}(x)$ and the pristine phase spectrum $\mathcal{P}(x)$ to obtain an augmented image.

Alternative approaches to augment images in the Fourier domain (Yang & Soatto, 2020; Xu et al., 2021) differ mostly in terms of how the function $g_\Lambda(\cdot)$ operates.

Across a set of synthetic source datasets, we make the important observation that synthetic images tend to have smaller variations in the high-frequency components of their amplitude spectrum than real images. Fig. 2 shows sample amplitude spectrums of images from different datasets (on the left) and shows the standard deviation of amplitude values for different frequency bands per-dataset (on the right).[1] We find that the variance in amplitude values for high frequency components is significantly higher for real as opposed to synthetic data (see Fig. 2 (b)). In Sec. A.4 of appendix, we show how this phenomenon is consistent across (1) several syn-to-real shifts and (2) fine-grained frequency band discretizations. This phenomenon is likely a consequence of how synthetic images are rendered. For instance, in VisDA-C, the synthetic images are viewpoint images of 3D object models (under different lighting conditions), so it's unlikely for them to be diverse in high-frequency details. For images from GTAV, synthetic renderings can lead to contributing factors such as low texture variations – for instance, "roads" (one of the head classes in semantic segmentation) in synthetic images likely have less high-frequency variation compared to real roads.[2] Consequentially, to generalize well to real data, we would like to ensure that our augmentation strategy exposes the model to more variations in the high-frequency components of the amplitude spectrum during training. Our intent is to ensure that the learned models are invariant to variations in high-frequency components – thereby avoiding overfitting to a specific syn-to-real shift.

**PASTA.** This is precisely where we come in. PASTA proposes perturbing the amplitude spectrums of the images in a manner that is proportional to the spatial frequencies, *i.e.*, higher frequencies are perturbed more compared to lower frequencies (see Fig. 3). For PASTA, we express $g_\Lambda(\cdot)$ as,

$$g_\Lambda(\mathcal{A}(x))[m,n] = \epsilon[m,n]\mathcal{A}(x)[m,n] \tag{4}$$

where $\epsilon[m,n] \sim \mathcal{N}(1, \sigma[m,n])$ and $\sigma[m,n] = \left(2\alpha\sqrt{\dfrac{m^2+n^2}{H^2+W^2}}\right)^k + \beta$ and $\Lambda = \{\alpha, k, \beta\}$ (5)

We ensure that the perturbation applied at spatial frequency $(m,n)$ has a direct dependence on $(m,n)$ (Eqn. 5). $\Lambda = \{\alpha, k, \beta\}$ are controllable hyper-parameters. $\beta$ ensures a baseline level of jitter applied to all frequencies and $\alpha, k$ govern how the perturbations grow with increasing frequencies. Note that setting either $\alpha = 0$ or $k = 0$ (removing the frequency dependence) results in a setting where the $\sigma[m,n]$ is the same across all $(m,n)$. In Sec. A.4 of appendix, we show quantitatively how applying PASTA increases the variance metric measured in Fig. 2 (b) for synthetic images across fine-grained frequency band discretizations.

## 4 EXPERIMENTS

We conduct our experiments across three tasks – semantic segmentation, object detection and object recognition, where we train on synthetic data so as to generalize to real world data.

### 4.1 DATASETS

**Semantic Segmentation.** For segmentation, we consider GTAV (Richter et al., 2016) as our synthetic source dataset. GTAV consists of $\sim$ 25k driving-scene images with 19 object classes. We consider Cityscapes (Cordts et al., 2016), BDD-100K (Yu et al., 2020) and Mapillary (Neuhold et al., 2017) as our real target datasets which contain $\sim$ 5k, $\sim$ 8k, and $\sim$ 25k finely annotated real-world driving / street view images respectively. The 19 classes in GTAV are compatible with those of Cityscapes, BDD-100K and Mapillary. We train all our models on the training split of the source synthetic dataset and evaluate on the validation splits of the real target datasets. We report performance in terms of mIoU (mean intersection over union).

**Object Detection.** For object detection, we consider Sim10K (Johnson-Roberson et al., 2016) as our synthetic source dataset. Sim10K consists of $\sim$ 10k images of street scenes obtained from GTAV with $\sim$ 59k bounding boxes for cars. We consider Cityscapes (Cordts et al., 2016) as our target

---

[1]In Fig. 2(b), for every image, upon obtaining the amplitude spectrum, we first take an element-wise logarithm. Then, for a particular frequency band (pre-defined), we compute the standard deviation of amplitude values within that band (across all the channels). Finally, we average these standard deviations across images to report the same in the bar plots.

[2]When PASTA is applied, we find that performance on "road" increases by a significant margin (see per-class generalization results in Sec. A.2 of appendix).

Table 1: **Synthetic-to-Real generalization results for semantic segmentation and object detection.** Tables 1a, 1b and 1c summarize syn-to-real generalization results for semantic segmentation (across 3 runs) on Cityscapes (C), BDD-100K (B) and Mapillary (M) when trained on GTAV (G). Table 1d summarizes syn-to-real generalization results for object detection on Cityscapes (C) when trained on Sim10K (S). $^*$ indicates numbers drawn from published manuscripts. $^\dagger$ indicates trained with downsampled $1024 \times 560$ images due to restricted compute. $k = 2$ and $\beta = 0.25$ for PASTA ($\alpha = 3$). $\beta = 0.5$ for PASTA ($\alpha = 0$). Rows in gray font use different base architectures and / or extra data for training and have been included primarily for completeness (drawn directly from published manuscripts). **Bold** indicates best and underline indicates second best.

(a) SemSeg G→{C, B, M} (R-50, DeepLabv3+).

| Method | C | B | M | Avg. |
|---|---|---|---|---|
| 1 Baseline$^*$ | 28.95 | 25.14 | 28.18 | 27.42 |
| 2 + RandAug | 31.89 | 38.28 | 34.54 | 34.54$_{\pm 0.57}$ |
| 3 + PASTA ($\alpha = 0$) | 41.65 | 34.75 | 43.37 | 39.92$_{\pm 0.30}$ |
| 4 + PASTA ($\alpha = 3$) | **44.12** | **40.19** | **47.11** | **43.81$_{\pm 0.74}$** |
| 5 IBN-Net$^*$ | 33.85 | 32.30 | 37.75 | 34.63 |
| 6 + PASTA ($\alpha = 3$) | **41.90** | **41.46** | **45.88** | **43.08$_{\pm 0.37}$** |
| 7 ISW$^*$ | 36.58 | 35.20 | 40.33 | 37.37 |
| 8 + PASTA ($\alpha = 3$) | **42.13** | **40.95** | **45.67** | **42.91$_{\pm 0.27}$** |
| 9 DRPC | 37.42 | 32.14 | 34.12 | 34.56 |
| 10 WEDGE | 38.15 | 36.14 | 43.21 | 39.17 |
| 11 ASG | 31.89 | N/A | N/A | N/A |
| 12 CSG | 35.27 | N/A | N/A | N/A |
| 13 WildNet | 44.62 | 38.42 | 46.09 | 43.04 |

(b) SemSeg G→{C, B, M} (R-101, DeepLabv3+).

| Method | C | B | M | Avg. |
|---|---|---|---|---|
| 1 Baseline$^*$ | 32.97 | 30.77 | 30.68 | 31.47 |
| 2 Baseline$^\dagger$ | 29.36 | 23.61 | 28.14 | 27.04$_{\pm 1.86}$ |
| 3 + PASTA ($\alpha = 3$)$^\dagger$ | **41.54** | **39.43** | **45.05** | **42.01$_{\pm 0.26}$** |
| 4 IBN-Net$^*$ | 37.37 | 34.21 | 36.81 | 36.13 |
| 5 IBN-Net$^\dagger$ | 30.85 | 32.94 | 34.77 | 32.35$_{\pm 0.70}$ |
| 6 + PASTA ($\alpha = 3$)$^\dagger$ | **40.74** | **41.62** | **44.71** | **42.36$_{\pm 0.58}$** |
| 7 ISW$^*$ | 37.20 | 33.36 | 35.57 | 35.58 |
| 8 ISW$^\dagger$ | 31.74 | 31.90 | 34.94 | 32.86$_{\pm 0.48}$ |
| 9 + PASTA ($\alpha = 3$)$^\dagger$ | **40.93** | **41.68** | **44.40** | **42.34$_{\pm 0.86}$** |
| 10 DRPC | 42.53 | 38.72 | 38.05 | 39.77 |
| 11 WEDGE | 43.60 | 41.62 | 48.82 | 44.68 |
| 12 ASG | 32.79 | N/A | N/A | N/A |
| 13 CSG | 38.88 | N/A | N/A | N/A |
| 14 FSDR | 44.80 | 41.20 | 43.40 | 43.13 |
| 15 WildNet | 45.79 | 41.73 | 47.08 | 44.87 |

(c) SemSeg G→{C, B, M} (MNv2, DeepLabv3+).

| Method | C | B | M | Avg. |
|---|---|---|---|---|
| 1 Baseline$^*$ | 25.92 | 25.73 | 26.45 | 26.03 |
| 2 Baseline$^\dagger$ | 24.27 | 23.30 | 23.07 | 23.55$_{\pm 0.47}$ |
| 3 + PASTA ($\alpha = 3$)$^\dagger$ | **37.76** | **36.32** | **39.07** | **37.71$_{\pm 0.54}$** |
| 4 IBN-Net$^*$ | 30.14 | 27.66 | 27.07 | 28.29 |
| 5 IBN-Net$^\dagger$ | 29.13 | 26.29 | 28.34 | 27.92$_{\pm 1.76}$ |
| 6 + PASTA ($\alpha = 3$)$^\dagger$ | **35.37** | **36.15** | **38.60** | **36.71$_{\pm 0.65}$** |
| 7 ISW$^*$ | 30.86 | 30.05 | 30.67 | 30.53 |
| 8 ISW$^\dagger$ | 30.11 | 27.20 | 27.54 | 28.28$_{\pm 1.65}$ |
| 9 + PASTA ($\alpha = 3$)$^\dagger$ | **35.15** | **36.49** | **38.85** | **36.83$_{\pm 0.40}$** |

(d) ObjDet S→C (R-50 & R-101, Faster-RCNN).

| Method | R-50 | R-101 |
|---|---|---|
| Generalization | | |
| 1 Baseline$^*$ | N/A | 41.8 |
| 2 Baseline | 39.4 | 43.3 |
| 3 + PD | 51.5 | 52.2 |
| 4 + RandAug | 52.8 | **57.2** |
| 5 + PASTA ($\alpha = 0$) | 52.9 | 53.9 |
| 6 + PASTA ($\alpha = 3$) | 56.3 | 55.2 |
| 7 + PD + PASTA ($\alpha = 0$) | 57.0 | 57.1 |
| 8 + PD + PASTA ($\alpha = 3$) | **58.0** | 56.6 |
| Adaptation | | |
| 9 EPM | N/A | 51.20 |
| 10 ILLUME | N/A | 53.10 |
| 11 Faster-RCNN (Oracle) | N/A | 70.40 |

dataset. Following prior work (Khindkar et al., 2022), we (1) train on the entirety of Sim10K and report performance on the validation split of Cityscapes and (2) consider detecting instances of the class "car" across Sim10K and Cityscapes. We use mAP@50 (mean average precision at an IoU threshold of 0.5) to report performance.

**Object Recognition.** For object recognition, we consider the VisDA-2017 (Peng et al., 2017) image-classification benchmark. The source synthetic domain, consists of 3D renderings of 12 object categories from different angles and under different lighting conditions – resulting in a total of $\sim$ 152k synthetic images. The target real domain ("real" val split in VisDA-C) consists of $\sim$ 55k images of the 12 classes cropped from images from the COCO dataset. We split the source domain into an 80/20 train / val split (for checkpoint selection) and evaluate on the entirety of the target domain specified above. For comparisons with CSG on VisDA-C, we use the same experimental configurations as (Chen et al., 2021). We use accuracy as our evaluation metric.

### 4.2 MODELS, IMPLEMENTATION DETAILS AND BASELINES

**Models and Implementation Details.** For our semantic segmentation experiments, we consider the DeepLabv3+ (Chen et al., 2018a) architecture with backbones – ResNet-50 (R-50) (He et al., 2016), ResNet-101 (R-101) (He et al., 2016) and MobileNetv2 (MN-v2) (Sandler et al., 2018) (see Tables 1a, 1b and 1c). Unlike Table. 1a (R-50), for Tables. 1b (R-101) and 1c (MN-v2), we downsample source GTAV images to the resolution $1024 \times 560$ for faster training (due to limited computational resources). For our object detection experiments, we consider the Faster-RCNN (Ren et al., 2015) architecture with R-50 and R-101 backbones (see Table. 1d). For segmentation and detection, PASTA is applied with some consistent positional and photometric augmentations. For our object recognition experiments, we consider two sets of experiments – (1) where we train a baseline ResNet-50 CNN from scratch with different augmentation strategies (see Table 2) and (2) where we compare our proposed approach with (Chen et al., 2021) with a ResNet-101 backbone (see Table 3). We provide more details in Sec. A.1 of the appendix.

Table 2: **VisDA-C (ResNet-50) generalization.** Vanilla ResNet-50 CNN trained (3 runs) on the synthetic source data of VisDA-C is evaluated on the real (val split) target data of VisDA-C. $k = 3$ and $\beta = 0.875$. I.D. and O.O.D. are regular in and out-of-domain accuracies. O.O.D. (Bal.) is class balanced accuracy on out-of-domain data. **Bold** indicates best and underline indicates second best.

| Method | Accuracy | | |
|---|---|---|---|
| | I.D. | O.O.D | O.O.D (Bal.) |
| 1 Baseline | $93.87_{\pm 0.13}$ | $55.86_{\pm 0.79}$ | $49.77_{\pm 1.49}$ |
| 2+ RandAug | $90.83_{\pm 0.34}$ | $\mathbf{60.48}_{\pm 0.34}$ | $\underline{56.40}_{\pm 0.33}$ |
| 3+ PASTA ($\alpha = 0$) | $\underline{92.13}_{\pm 0.41}$ | $56.29_{\pm 1.27}$ | $51.61_{\pm 1.84}$ |
| 4+ PASTA ($\alpha = 4$) | $90.26_{\pm 0.47}$ | $\underline{60.20}_{\pm 0.71}$ | $\mathbf{56.91}_{\pm 0.53}$ |

**Baselines and points of comparison.** In addition to prior work, we compare PASTA with (1) RandAugment (Cubuk et al., 2020) and (2) the setting where $\alpha = 0$ in PASTA to assess the extent to which monotonic increase in $\sigma[m, n]$ makes a difference. For segmentation and detection, we consider only the pixel-level / photometric transforms in the RandAugment vocabulary. For our object recognition experiments, we consider the entire RandAugment vocabulary.

## 5 RESULTS AND FINDINGS

### 5.1 SYNTHETIC-TO-REAL GENERALIZATION RESULTS

Our semantic segmentation results are summarized in Tables. 1a, 1b and 1c, object detection results in Table. 1d and object recognition results in Tables. 2 and 3.

**PASTA consistently improves performance.** We observe that PASTA offers consistent improvements for all three tasks and the considered synthetic-to-real shifts.

– **Semantic Segmentation.** From Tables. 1a, 1b and 1c, we observe that PASTA significantly improves performance of a baseline model ($\sim$16%, $\sim$10% and $\sim$11% mIoU for R-50, R-101 and MN-v2 respectively) and outperforms RandAugment (Cubuk et al., 2020) by a significant margin – 9.3% for R-50 (Table. 1a rows 2,4).[3] Additionally, we find that PASTA makes the baseline DeepLabv3+ model improve over existing approaches that use either extra data, modeling components, or objectives by a significant margin. For instance, Baseline + PASTA rows across Tables. 1a, 1b and 1c outperforms IBN-Net, ISW, DRPC (Yue et al., 2019), ASG (Chen et al., 2020b) and CSG (Chen et al., 2021). For WEDGE (Kim et al., 2021), FSDR (Huang et al., 2021a) & WildNet (Lee et al., 2022) we find that Baseline + PASTA outperforms only for R-50 and not for R-101. We would like to note that DRPC, ASG, CSG, WEDGE, FSDR, WildNet use either more data (the entirety of GTAV or additional datasets) or different base architectures, making these comparisons unfair *to* PASTA. We would also like to note that for our experiments using R-101 and MN-v2, due to limited computational resources, we downsample GTAV images to a lower resolution while training. This slightly hurts PASTA performance relative to experiments run at the original resolution (e.g. performance of Table. 1a row 4 drops from $43.81$ to $41.90$ when downsampling during training). In Sec. A.2 of appendix, we also show that for the SYNTHIA $\rightarrow$ Real shift (1) PASTA provides strong improvements over the vanilla baseline ($+3.91\%$ absolute improvement) and (2) is competitive with RandAugment.

– **Object Detection.** For Object Detection (see Table. 1d), we compare PASTA to RandAugment and PhotoMetric Distortion (PD). PD randomly applies a sequence of operations to alter the contrast, brightness, hue, saturation and other photometric properties of the input image (details in Sec. A.1 of appendix). For both R-50 and R-101, we first note that PASTA (1) improves the performance of

---

[3]In Sec. A.2 of appendix, we show how PASTA improves across several classes for the considered shifts.

a baseline model ($\sim 17\%$ and $\sim 12\%$ for R-50 and R-101 respectively) and (2) improves over PD and RandAugment for R-50 and is competitive with RandAugment for R-101. More interestingly, for R-101, we find that not only does PASTA improve performance on real data but also significantly outperforms the state-of-the-art adaptive object detection method – ILLUME (Khindkar et al., 2022), a method that has access to target data at training time!

– **Object Recognition.** For object recognition (see Tables. 2 and 3), we find that while PASTA improves over a vanilla baseline, the offered improvements are competitive with that of RandAugment. From rows 1, 2, 4 in Table. 2, we can observe that while both RandAugment and PASTA outperform the Baseline, RandAugment and PASTA ($\alpha = 4$) are competitive when we look at best syn-to-real generalization performance. In Table. 3 (rows 8, 9, 10) we can observe that CSG is competitive with versions where RandAugment in CSG is replaced with PASTA.

**PASTA is complementary to existing generalization methods.** In addition to ensuring that a baseline model is competitive with or significantly improves over existing methods, PASTA can also complement to existing generalization methods. For semantic segmentation, from Tables. 1a, 1b and 1c, we find that applying PASTA to IBN-Net and ISW significantly improves performance (see IBN-Net, IBN-Net + PASTA and ISW, ISW + PASTA set of rows) – with improvements ranging from $\sim$6-9% mIoU for IBN-Net and $\sim$5-7% mIoU for ISW across different backbones. For object recognition, in Table. 3, we apply PASTA to CSG (Chen et al., 2021), one of the state-of-the-art generalization methods on VisDA-C. Since CSG inherently uses RandAugment, we consider both settings where PASTA is applied to CSG with and without RandAugment. In both cases, applying PASTA improves performance.

**PASTA vs Frequency-based Augmentation Strategies.** Closest to our work is FACT (Xu et al., 2021), where authors employ Amplitude Mixup (AM) and Amplitude Jitter (AJ) as augmentations in a broader training pipeline. On our experimental settings (semantic segmentation using a R-50 DeepLabv3+ baseline) we find that PASTA outperforms AM and AJ – 41.90% (PASTA) vs 39.70% (AM) vs 30.70% (AJ) mIoU on real datasets (single run, trained at $1024 \times 560$). Since FACT was designed specifically for multi-source domain generalization settings (not necessarily in a syn-to-real context), we also compare directly with FACT on PACS (a multi-source DG setup for object recognition) by replacing the augmentation pipeline in FACT with PASTA. We find that FACT-PASTA outperforms FACT-Vanilla – 87.97% vs 87.10% average leave-one-out domain accuracy. We further note that PASTA is also competitive (within 1%) with FSDR (Huang et al., 2021a), another frequency space domain randomization method, which uses extra data in its training pipeline (see Table. 1b).[4]

Table 3: **VisDA-C (ResNet-101) generalization.** We apply PASTA to CSG (Chen et al., 2021). Since CSG inherently uses RandAug, we also report results with and without the use of RandAug when PASTA is applied. * indicates drawn directly from published manuscripts. We report class-balanced accuracy on the real (val split) target data of VisDA-C. $k = 1$ and $\beta = 0.5$ for PASTA. Results are reported across 3 runs. **Bold** indicates best and underline indicates second best.

| Method | Accuracy |
|---|---|
| 1 Oracle (IN-1k)* | 53.30 |
| 2 Baseline (Syn. Training)* | 49.30 |
| 3 Weight $\ell 2$ Distance* | 56.40 |
| 4 Synaptic Intelligence* | 57.60 |
| 5 Feature $\ell 2$ Distance* | 57.10 |
| 6 ASG* | 61.10 |
| 7 CSG* | 64.05 |
| 8 CSG | $63.84_{\pm 0.29}$ |
| w/o RandAug | |
| 9 CSG + PASTA ($\alpha = 0$) | $64.05_{\pm 0.69}$ |
| 10 CSG + PASTA ($\alpha = 10$) | $64.29_{\pm 0.56}$ |
| w RandAug | |
| 11 CSG + PASTA ($\alpha = 0$) | $\underline{65.38}_{\pm 0.80}$ |
| 12 CSG + PASTA ($\alpha = 10$) | $\mathbf{65.86}_{\pm 1.13}$ |

### 5.2 Analyzing PASTA

**Sensitivity of PASTA to $\alpha$, $k$ and $\beta$.** We discuss how sensitive generalization performance is to the choice of $\alpha$, $k$ and $\beta$ in PASTA and attempt to provide general guidelines and pitfalls for appropriately selecting the same. We note the following:

– **Same choice of $\alpha$, $k$ and $\beta$ leads to consistent improvements across multiple tasks (object detection and semantic segmentation).** We find that the same set of values, $\alpha \in \{0, 3\}, k = 2, \beta \in \{0.25, 0.5\}$, offer significant improvements across (1) both the tasks of detection and segmentation, (2) different backbones: R-50 and R-101 and (3) different shifts: $\{$GTAV, Sim10K$\} \rightarrow \{$Cityscapes, BDD100K, Mapillary$\}$.

– **Optimal values of $\alpha$, $k$ and $\beta$ are different across different kinds of shifts.** We find that values of hyper-parameters that lead to improved performance differ across different synthetic source

---

[4]We discuss these experiments in Sec. A.3 of appendix.

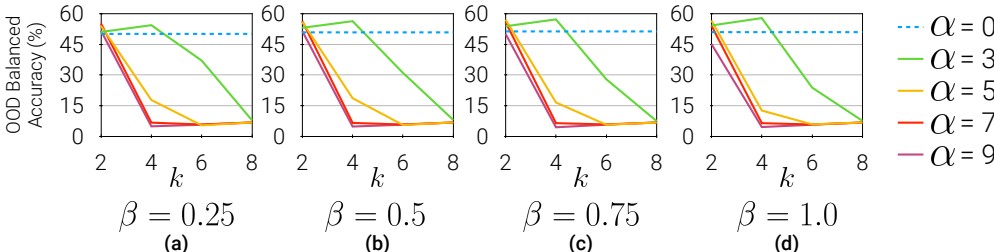

Figure 4: **Ablating $\alpha$, $\beta$, $k$ in PASTA.** We train a vanilla ResNet-50 on VisDA-C source by applying PASTA and varying $\alpha$, $k$ and $\beta$ within the sets $\{0, 3, 5, 7, 9\}$, $\{2, 4, 6, 8\}$ and $\{0.25, 0.5, 0.75, 1.0\}$ respectively. The trained models are evaluated on the target (validation) real data with class-balanced accuracy as the metric.

datasets {GTAV, Sim10K} and VisDA-C (Tables. 1a, 1b, 1c, 1d vs 2, 3). This is not entirely surprising since VisDA-C and {GTAV, Sim10K} images differ significantly – GTAV includes synthetic street views and VisDA-C includes different viewpoints of 3D CAD models of objects.

– **Appropriate selection of $\alpha$ and $k$ is more important compared to $\beta$.** In Fig. 4 we evaluate R-50 trained on synthetic VisDA-C source on the real (validation) target split for different values of $\alpha$, $k$ and $\beta$ and report generalization performance. More than anything, we find that performance is quite sensitive to the choice of $k$ – performance increases slightly with increasing $k$ and then drops significantly. This holds across all values of $\beta$. Furthermore, increasing $\alpha$ for a value of $k$ results in significant drops. Qualitatively, we observe that for high $\alpha$ and $k$ we get augmented images have their semantic content significantly occluded, thereby, resulting in poor generalization.

**Is PASTA complementary to other augmentation strategies?** We first check if PASTA is complementary to RandAugment. To assess this, we conduct experiments with a vanilla R-50 CNN on VisDA-C by using both PASTA and RandAugment. We find that, in terms of absolute values, PASTA + RandAugment outperforms RandAugment but not PASTA, but all performances are within 1% of each other. We notice that combining the best PASTA setting with the best RandAugment one leads to very strong augmentations and further note that reducing the strength of PASTA in the combination leads to improved results. On semantic segmentation, we find that a simple combination of just RGB gaussian noise and color jitter itself leads to significant improvements in generalization (leading to a mIoU of almost 40.96%; single run, trained at $1024 \times 560$) – with gaussian noise likely adding more high-frequency components and color jitter likely perturbing low-frequency components. PASTA on the other hand is a more fine-grained method to perturb all frequencies in a structured manner, which is likely why it outperforms this combination.

To summarize, across our set of extensive experiments, we observe that PASTA serves as a simple and effective augmentation strategy that – (1) ensures a baseline method is competitive with existing generalization approaches, (2) often improves performance over existing generalization methods and (3) is complementary to existing generalization methods.

**Limitations.** By relying on the observation that synthetic images can be lacking in high-frequency variations, even though PASTA leads to strong generalization results, akin to any augmentation strategy, it is not without it's limitations. Firstly, it is important to note that PASTA has only been tested on the syn-to-real generalization settings explored in this work and it is an important step for future work to assess the extent to which improvements offered by PASTA translate to more such settings. Secondly, as observed in Sec. 5.2, PASTA is quite sensitive to the choice of $k$, with a small subset of values of $\alpha$, $k$ and $\beta$ defining beneficial levels of augmentations.

## 6  CONCLUSION

We propose Proportional Amplitude Spectrum Training Augmentation (PASTA), an augmentation strategy for synthetic-to-real generalization. PASTA is motivated by the observation that the amplitude spectra are less diverse in synthetic than real data, especially for high-frequency components. Thus, PASTA augments synthetic data by perturbing the amplitude spectra, with magnitudes increasing for higher frequencies. We show that PASTA offers strong out-of-the-box generalization performance on semantic segmentation, object detection, and object classification tasks. The strong performance of PASTA holds true alone (i.e., training with ERM using PASTA augmented images) or together with alternative generalization/augmentation algorithms. We would like to emphasize that the strength of PASTA lies in its simplicity and effectiveness, offering strong improvements despite not using extra modeling components, objectives, or extra data. We hope that future research endeavors in syn-to-real generalization take domain randomization techniques like PASTA into account.

## 7 ETHICS STATEMENT

While simulation offers the promise of having more diverse synthetic training data, this is predicated on the assumption that simulators will be designed to produce the diversity of data needed to represent the world. Most existing work on developing simulated visuals focus on diversity such as weather, lighting, and sensor choice. In order for syn-to-real transfer to be effective for the world population we will also need to consider design and collection of synthetic data that adequately represents all sub-populations.

## 8 REPRODUCIBILITY STATEMENT

For all our experiments surrounding PASTA and associated baselines, we provide details surrounding the datasets and training, validation and test splits in Sec. 4 of the main paper and Sec. A.1 of the appendix. Sec. A.1 also provides details surrounding the choice of hyper-parameters, optimization, and model selection criteria for all the three tasks of semantic segmentation, object detection, and object recognition. Sec. A.6 of the appendix provides details about the frameworks and base code repositories used for our experiments. We will release our code and data upon acceptance.

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

# A APPENDIX

This appendix is organized as follows. In Sec. A.1, we first expand on implementation and training details from the main paper. Then, in Sec. A.2, we provide per-class synthetic-to-real generalization results (see Sec. 5.1 of the main paper). Sec. A.3 includes additional discussions surrounding different aspects of PASTA. Sec. A.4 includes additional analysis of the amplitudes across multiple frequency bands and datasets. Next, Sec. A.5 contains more qualitative examples of PASTA augmentations and predictions for semantic segmentation. Finally, Sec. A.6 summarizes the licenses associated with different assets used in our experiments.

## A.1 IMPLEMENTATION AND TRAINING DETAILS

In this section, we outline our training and implementation details for each of the three tasks – Semantic Segmentation, Object Detection, and Object Recognition. We also summarize these details in Tables. 4a, 4b, and 4c.

**Semantic Segmentation (see Table. 4a).** As stated in Sec. 4.2 of the main paper, for our semantic segmentation experiments, we consider the DeepLabv3+ (Chen et al., 2018a) architecture with backbones – ResNet-50 (R-50) (He et al., 2016), ResNet-101 (R-101) (He et al., 2016) and MobileNetv2 (MN-v2) (Sandler et al., 2018) (see Tables. 1a, 1b, and 1c in the main paper). In our experiments using R-101 and MN-v2, we resize the source GTAV images to the resolution $1024 \times 560$ for faster training under limited computational resources. We adopt the hyper-parameter (and distributed training) settings used by (Choi et al., 2021) for training. We train ResNet-50, ResNet-101 and MobileNetv2 based models across 4, 4 and 2 GPUs respectively in a distributed manner (similar to (Choi et al., 2021)). This includes the use of SGD as an optimizer with an initial learning rate of $10^{-2}$ and a momentum of 0.9. Similar to (Choi et al., 2021), we also use a polynomial learning rate schedule (Liu et al., 2015) with a power of 0.9. Our models are initialized with ImageNet (Krizhevsky et al., 2012) pre-trained weights. We train all our models for 40k iterations with a batch size of 16 for GTAV. Our segmentation models are trained on the train split of GTAV and evaluated on the validation splits of the target datasets (Cityscapes, BDD-100K and Mapillary). For segmentation, PASTA is applied with a base set of positional and photometric augmentations (PASTA first and then the base augmentations) – gaussian blur, color jitter, random crop, random horizontal flip and random scaling are used. We use Pytorch (Paszke et al., 2019) implementations of these augmentations. For RandAugment (Cubuk et al., 2020), we only consider the vocabulary of photometric augmentations for segmentation. We conduct ablations (within computational constraints) for the best performing RandAugment setting using R-50 for syn-to-real generalization and find that best performance is achieved when 8 augmentations are sampled from the vocabulary for application at the highest severity level (30). Whenever we train a prior generalization approach, ISW (Choi et al., 2021) or IBN-Net (Pan et al., 2018), we follow the same set of hyper-parameter configurations as (Choi et al., 2021). All models are trained across 3 random seeds.

**Object Detection (see Table. 4b).** For object detection, we consider the Faster-RCNN (Ren et al., 2015) architecture with ResNet-50 and ResNet-101 backbones (see Table. 1d in the main paper). We train on the entirety of Sim10K Johnson-Roberson et al. (2016) (source dataset) for 10k iterations and pick the last checkpoint for Cityscapes (target dataset) evaluation. We use SGD with momentum as our optimizer with an initial learning rate of $10^{-2}$ (adjusted according to a step learning rate schedule) and a batch size of 32. Our models are initialized with ImageNet (Krizhevsky et al., 2012) pre-trained weights. All models are trained on 4 GPUs. For detection, we also compare PASTA against RandAugment (Cubuk et al., 2020) and Photometric Distortion (PD). The sequence of operations in PD to augment input images are – randomized brightness, randomized contrast, RGB→HSV conversion, randomized saturation & hue changes, HSV→RGB conversion, randomized contrast, and randomized channel swap.

**Object Recognition (see Table. 4c).** As stated earlier, for object recognition, we consider two sets of experiments (see Tables. 2 and 3 in the main paper) – (1) where we train a baseline ResNet-50 CNN from scratch with different augmentation strategies (see Table. 2, main paper) and (2) where we check if PASTA is complementary to CSG (Chen et al., 2021), when trained with a ResNet-101 backbone (see Table. 3, main paper). For (1), we split the synthetic source data into a training and validation split in an 80/20 ratio for checkpoint selection. We use SGD without momentum as an optimizer with a learning rate of $10^{-4}$, weight decay of $10^{-5}$ and batch size of 64. We train these models for 10 epochs. Our models are initialized with ImageNet (Krizhevsky et al., 2012) pre-trained weights. We use Pytorch (Paszke et al., 2019) implementations of color jitter, random resized crop and random horizontal flip as the base set of augmentations. For (2), we follow the same hyper-parameter

Table 4: **Implementation & Optimization Details.** We summarize details surrounding dataset, training, optimization and model selection criteria for our semantic segmentation, object detection and object recognition experiments.

(a) Semantic Segmentation Training

| Config | Value |
|---|---|
| Source Data | GTAV (Train Split) |
| Target Data | Cityscapes (Val Split) |
| | BDD-100K (Val Split) |
| | Mapillary (Val Split) |
| Segmentation Architecture | DeepLabv3+ (Chen et al., 2018a) |
| CNN Backbones | ResNet-50 (R-50) (He et al., 2016) |
| | ResNet-101 (R-101) (He et al., 2016) |
| | MobileNetv2 (Mn-v2) (Sandler et al., 2018) |
| Training Resolution | Original GTAV resolution for R-50 |
| | $1024 \times 560$ for R-101 & Mn-v2 |
| Optimizer | SGD |
| Initial Learning Rate | $10^{-2}$ |
| Learning Rate Schedule | Polynomial LR (exp 0.9) |
| Initialization | Imagenet Pre-trained Weights (Krizhevsky et al., 2012) |
| Iterations | 40000 |
| Batch Size | 16 |
| Augmentations w/ PASTA | Gaussian Blur, Color Jitter, Random Crop |
| | Random Horizontal Flip, Random Scaling |
| Model Selection Criteria | In-domain (Source) Validation performance |

(b) Object Detection Training

| Config | Value |
|---|---|
| Source Data | Sim10K |
| Target Data | Cityscapes (Val Split) |
| Segmentation Architecture | Faster-RCNN (Ren et al., 2015) |
| CNN Backbones | ResNet-50 (R-50) (He et al., 2016) |
| | ResNet-101 (R-101) (He et al., 2016) |
| Training Resolution | Original Sim10K resolution for R-50, R-101 |
| Optimizer | SGD (momentum = 0.9) |
| Initial Learning Rate | $10^{-2}$ |
| Learning Rate Schedule | Step-LR, Warmup 500 iterations, Warmup Ratio 0.001 |
| | Steps 6000 & 8000 iterations |
| Initialization | Imagenet Pre-trained Weights (Krizhevsky et al., 2012) |
| Iterations | 10000 |
| Batch Size | 32 |
| Augmentations w/ PASTA | Resize, Random Flip |
| Model Selection Criteria | Checkpoint at 10k iterations |

(c) Object Recognition Training

| Config | Value (for Table. 2 main paper) | Value (for Table. 3 main paper) |
|---|---|---|
| Source Data | VisDA-C Synthetic | VisDA-C Synthetic |
| Target Data | VisDA-C Real | VisDA-C Real |
| Backbone | ResNet-50 (R-50) (He et al., 2016) | ResNet-101 (R-101) (He et al., 2016) |
| Optimizer | SGD w/o momentum | SGD w/ momentum (0.9) |
| Initial Learning Rate | $10^{-4}$ | $10^{-4}$ |
| Weight Decay | $10^{-5}$ | $5 \times 10^{-4}$ |
| Initialization | Imagenet (Krizhevsky et al., 2012) | Imagenet (Krizhevsky et al., 2012) |
| Epochs | 10 | 30 |
| Batch Size | 64 | 32 |
| Augmentations w/ PASTA | Color Jitter, Random Resized Crop | RandAugment (Cubuk et al., 2020) |
| | Random Horizontal Flip | |
| Model Selection Criteria | Best In-domain Val Performance | Best In-domain Val Performance |

Table 5: **GTA5→Cityscapes per-class generalization results.** Per-class IoU comparisons for syn-to-real generalization results when semantic segmentation models trained on GTAV are evaluated on Cityscapes. Base model involved is DeepLabv3+ (R-50 backbone). Results are reported across 3 runs. * indicates drawn directly from published manuscripts. $k = 2$ and $\beta = 0.25$ for PASTA ($\alpha = 3$). $\beta = 0.5$ for PASTA ($\alpha = 0$). Class headers are in decreasing order of pixel frequency.

| Method | road | building | vegetation | car | sidewalk | sky | pole | person | terrain | fence | wall | bicycle | sign | bus | truck | rider | light | train | motorcycle | mIoU |
|---|---|---|---|---|---|---|---|---|---|---|---|---|---|---|---|---|---|---|---|---|
| 1 Baseline | 45.1 | 56.8 | 80.9 | 61.0 | 23.1 | 38.9 | 23.9 | 58.2 | 24.3 | 16.3 | 16.6 | 13.4 | 7.3 | 20.0 | 17.4 | 1.2 | 30.0 | 7.2 | 8.5 | 29.0 |
| 2 +RandAug | 58.5 | 56.3 | 77.3 | 83.7 | 30.3 | 45.1 | 27.3 | 57.8 | 20.6 | 20.9 | 11.4 | 16.9 | 9.7 | 20.3 | 15.0 | 2.4 | 28.1 | 14.0 | 10.3 | 31.9 |
| 3 +PASTA ($\alpha = 0$) | 77.1 | 76.3 | 85.5 | 82.3 | 35.2 | 72.9 | **33.1** | 65.2 | **31.6** | 25.7 | 25.4 | **22.8** | 19.8 | 23.2 | **33.8** | 2.8 | **33.8** | 21.0 | 24.0 | 41.7 |
| 4 +PASTA ($\alpha = 3$) | **84.1** | **80.5** | **85.8** | **85.9** | **40.1** | **81.8** | 31.9 | **66.0** | 31.4 | **28.1** | **29.0** | 21.8 | **28.5** | **24.5** | 28.7 | **7.0** | 32.9 | **23.4** | **27.2** | **44.1** |
| 5 IBN-Net | 51.3 | 59.7 | 85.0 | 76.7 | 24.1 | 67.8 | 23.0 | 60.6 | **40.6** | **25.9** | 14.1 | 15.7 | 10.1 | 23.7 | 16.3 | 0.8 | 30.9 | 4.9 | 11.9 | 33.9 |
| 6 +PASTA ($\alpha = 3$) | **78.1** | **79.5** | **85.8** | **84.5** | **31.7** | **80.1** | **32.2** | **63.4** | 38.8 | 21.7 | **28.0** | **18.2** | **22.6** | **26.4** | **29.0** | **2.8** | **34.0** | **16.5** | **22.9** | **41.9** |
| 7 ISW | 60.5 | 65.4 | 85.4 | 82.7 | 25.5 | 70.3 | 25.8 | 61.9 | 38.5 | **23.7** | 21.6 | 15.5 | 12.2 | 25.4 | 21.1 | 0.0 | 33.3 | 9.3 | 16.8 | 36.6 |
| 8 +PASTA ($\alpha = 3$) | **76.6** | **78.4** | **85.6** | **83.7** | **32.5** | **83.1** | **33.1** | **63.4** | **40.4** | 23.6 | **27.3** | **17.4** | **22.3** | **25.7** | **30.1** | **3.3** | **35.9** | **18.2** | **19.9** | **42.1** |

configurations adopted by (Chen et al., 2021). This includes the use of an SGD (with momentum 0.9) optimizer with a learning rate of $10^{-4}$, weight decay of $5 \times 10^{-4}$ and a batch size of 32. These models are trained for 30 epochs. CSG (Chen et al., 2021) also uses RandAugment (Cubuk et al., 2020) as an augmentation. For both (1) and (2), models are trained on single GPUs across 3 random seeds.

### A.2 SYNTHETIC-TO-REAL GENERALIZATION RESULTS

**Overall GTAV→Real Generalization Results.** We first note that using RandAugment (Cubuk et al., 2020) for the Baseline DeepLabv3+ model with ResNet-101 and MobileNetv2 backbones leads to overall syn-to-real generalization performances of 38.28±0.44 (PASTA performance being 42.01±0.26) and 30.47±1.22 (PASTA performance being 37.71±0.54) respectively. Overall, across MNv2, R-50 and R-101, we observe that the gap between RandAugment and PASTA reduces as the number of parameters in the backbone increase. In Tables. 1a, 1b and 1c of the main paper, we present overall synthetic-to-real generalization results when models trained on GTAV are evaluated on Cityscapes, BDD-100K and Mapillary. As stated in Sec. 5.1 of the main paper, for ResNet-50 and ResNet-101 (in Tables. 1a and 1b of the main paper), including comparisons with DRPC (Yue et al., 2019), ASG (Chen et al., 2020b), CSG (Chen et al., 2021), WEDGE (Kim et al., 2021), FSDR (Huang et al., 2021a) and WildNet (Lee et al., 2022) is not entirely fair *to* PASTA since these approaches use either more data or different base architectures for training. For instance, WEDGE (Kim et al., 2021) and CSG (Chen et al., 2021) use DeepLabv2, ASG (Chen et al., 2020b) uses FCNs, DRPC (Yue et al., 2019) uses the entirety of GTAV (not just the training split) and WEDGE uses ~5k extra Flickr images in it's overall pipeline. FSDR (Huang et al., 2021a) is another such approach that, when trained with a ResNet-101 backbone, achieves overall mIoU of 43.13% (within 1% of Baseline / IBN-Net / ISW + PASTA) on real datasets. However, FSDR also uses FCNs and the the entirety of GTAV for training. FSDR (Huang et al., 2021a) and WildNet (Lee et al., 2022) also use extra ImageNet (Krizhevsky et al., 2012) images for stylization / randomization. For FSDR, the first step in the pipeline also requires access to SYNTHIA (Ros et al., 2016).

**Per-class GTAV→Real Generalization Results.** Tables 5, 6 and 7 include per-class synthetic-to-real generalization results when a DeepLabv3+ (R-50 backbone) model trained on GTAV is evaluated on Cityscapes, BDD-100K and Mapillary respectively. We briefly discuss in the main paper how PASTA ($\alpha \geq 0$) improves performance across several classes for GTAV→Real. For GTAV→Cityscapes (see Table. 5), we find that Baseline + PASTA consistently improves over Baseline and RandAugment. For IBN-Net and ISW in this setting, we observe consistent improvements (except for the classes *terrain* and *fence*). For GTAV→BDD-100K (see Table. 6), we find that for the Baseline, while PASTA outperforms RandAugment on the majority of classes, both are fairly competitive and outperform the vanilla Baseline approach. For IBN-Net and ISW, PASTA ($\alpha = 3$) almost always outperforms the vanilla approaches (except for the class *wall*). For GTAV→Mapillary (see Table. 7), for Baseline, we find that PASTA ($\alpha \geq 0$) outperforms the vanilla approach and RandAugment. For IBN-Net and ISW, PASTA ($\alpha = 3$) outperforms the vanilla approaches with the exception of the classes *train* and *fence*.

**Overall SYNTHIA→Real Generalization Results.** We conducted additonal experiments using SYNTHIA (Ros et al., 2016) as the source domain and Cityscapes, BDD-100K and Mapillary as

Table 6: **GTA5→BDD-100K per-class generalization results.** Per-class IoU comparisons for syn-to-real generalization results when semantic segmentation models trained on GTAV are evaluated on BDD-100K. Base model involved is DeepLabv3+ (R-50 backbone). Results are reported across 3 runs. * indicates drawn directly from published manuscripts. $k = 2$ and $\beta = 0.25$ for PASTA ($\alpha = 3$). $\beta = 0.5$ for PASTA ($\alpha = 0$). Class headers are in decreasing order of pixel frequency.

| Method | road | sky | building | vegetation | car | sidewalk | fence | terrain | truck | pole | bus | wall | sign | person | light | bicycle | motorcycle | rider | train | mIoU |
|---|---|---|---|---|---|---|---|---|---|---|---|---|---|---|---|---|---|---|---|---|
| 1 Baseline | 48.2 | 32.3 | 34.3 | 58.2 | 67.3 | 23.0 | 19.8 | 21.4 | 11.3 | 28.4 | 5.6 | 3.4 | 18.1 | 43.9 | 30.2 | 11.1 | 16.0 | 5.1 | 0.0 | 25.1 |
| 2 +RandAug | 75.4 | 82.8 | 67.7 | **74.7** | 74.1 | **39.3** | **32.7** | **26.6** | 22.7 | **37.0** | **16.5** | 5.0 | 23.9 | 51.7 | 35.8 | 12.0 | 29.3 | **20.2** | 0.0 | 38.3 |
| 3 +PASTA ($\alpha = 0$) | 73.6 | 65.9 | 50.6 | 67.7 | 78.5 | 37.2 | 28.8 | 19.1 | 20.5 | 34.1 | 6.4 | 4.2 | 25.4 | 52.1 | 35.8 | 10.4 | **38.5** | 11.5 | 0.0 | 34.8 |
| 4 +PASTA ($\alpha = 3$) | **86.0** | **86.6** | **74.8** | 72.7 | **82.8** | 38.4 | 31.2 | 24.9 | **23.7** | 34.8 | 6.4 | **11.2** | **26.2** | **55.1** | **37.0** | 13.3 | 38.3 | 19.9 | 0.0 | **40.2** |
| 5 IBN-Net | 68.9 | 66.9 | 56.7 | 66.6 | 70.3 | 28.8 | 21.4 | 22.1 | 12.8 | 31.9 | 7.2 | 6.0 | 21.7 | 50.2 | 35.0 | 18.1 | 23.2 | 5.8 | 0.0 | 32.3 |
| 6 +PASTA ($\alpha = 3$) | **86.1** | **87.6** | **74.9** | 72.3 | 82.3 | 36.6 | **30.6** | 26.2 | 25.3 | 37.1 | 10.8 | **13.2** | 25.5 | 56.0 | 36.8 | 21.4 | 38.9 | 26.0 | 0.0 | 41.5 |
| 7 ISW | 74.9 | 77.4 | 65.2 | 69.0 | 72.4 | 30.4 | 22.6 | 26.2 | 16.2 | 34.9 | 6.1 | **11.5** | 22.2 | 50.3 | 36.9 | 11.4 | 31.3 | 10.0 | 0.0 | 35.2 |
| 8 +PASTA ($\alpha = 3$) | **86.5** | **87.9** | 74.0 | 73.0 | 83.2 | 37.7 | 28.6 | 28.1 | 23.4 | 37.2 | 7.8 | 11.3 | 25.0 | 55.1 | 37.8 | 23.6 | 35.5 | 22.4 | 0.0 | 41.0 |

Table 7: **GTA5→Mapillary per-class generalization results.** Per-class IoU comparisons for syn-to-real generalization results when semantic segmentation models trained on GTAV are evaluated on Mapillary. Base model involved is DeepLabv3+ (R-50 backbone). Results are reported across 3 runs. * indicates drawn directly from published manuscripts. $k = 2$ and $\beta = 0.25$ for PASTA ($\alpha = 3$). $\beta = 0.5$ for PASTA ($\alpha = 0$). Class headers are in decreasing order of pixel frequency.

| Method | sky | road | vegetation | building | sidewalk | car | fence | pole | terrain | wall | sign | truck | person | bus | light | bicycle | rider | motorcycle | train | mIoU |
|---|---|---|---|---|---|---|---|---|---|---|---|---|---|---|---|---|---|---|---|---|
| 1 Baseline | 42.2 | 46.8 | 64.9 | 33.5 | 24.9 | 72.3 | 14.4 | 27.7 | 23.8 | 6.7 | 8.5 | 23.7 | 53.7 | 7.0 | 35.8 | 18.4 | 4.9 | 15.5 | 10.8 | 28.2 |
| 2 +RandAug | 51.7 | 59.6 | 75.5 | 39.5 | 33.9 | 81.3 | 22.6 | 37.2 | 24.6 | 4.2 | 32.4 | 31.2 | 56.5 | 13.4 | 36.2 | 18.0 | 11.9 | 21.4 | 5.0 | 34.5 |
| 3 +PASTA ($\alpha = 0$) | 88.4 | 76.5 | 73.7 | 65.5 | 36.1 | 82.3 | 22.7 | 38.9 | 32.9 | 14.5 | 49.9 | **35.8** | 60.0 | 17.0 | **42.6** | 27.3 | 16.8 | 31.8 | 11.2 | 43.4 |
| 4 +PASTA ($\alpha = 3$) | **93.3** | **83.0** | 76.3 | 76.9 | 40.2 | 83.3 | 27.1 | 40.9 | 37.1 | 19.2 | 50.4 | 35.2 | **63.3** | 19.5 | 41.4 | **29.4** | 25.2 | 38.1 | 15.2 | **47.1** |
| 5 IBN-Net | 82.0 | 66.4 | 73.5 | 57.1 | 32.9 | 73.1 | 24.9 | 31.5 | 28.4 | 10.5 | 38.9 | 30.7 | 56.4 | 16.0 | 38.0 | 18.6 | 9.1 | 16.6 | **12.6** | 37.7 |
| 6 +PASTA ($\alpha = 3$) | **94.4** | 81.7 | 76.1 | 76.9 | 40.4 | 80.8 | 27.1 | 40.3 | 38.7 | 19.0 | 43.2 | 38.0 | 62.0 | 20.5 | 39.3 | 25.7 | 23.6 | 31.6 | 12.4 | 45.9 |
| 7 ISW | 88.2 | 74.8 | 74.3 | 66.1 | 36.2 | 78.7 | **26.0** | 35.4 | 30.2 | 15.2 | 36.6 | 33.3 | 58.6 | 14.4 | 37.9 | 17.8 | 11.1 | 20.4 | 11.0 | 40.3 |
| 8 +PASTA ($\alpha = 3$) | **94.6** | 82.9 | 76.7 | 76.0 | 41.9 | 81.8 | 25.8 | 40.4 | 40.9 | 18.8 | 43.1 | 34.1 | 61.6 | 19.9 | 40.2 | 24.5 | 22.3 | 30.5 | 11.6 | 45.7 |

the target domains with the same set of hyper-parameters for PASTA ($\alpha = 3, k = 2, \beta = 0.25$). For a baseline DeepLabv3+ model (R-101), we find that PASTA – (1) provides strong improvements over the vanilla baseline (31.77% mIoU, +3.91% absolute improvement) and (2) is competitive with RandAugment (32.30% mIoU). More generally, we find that syn-to-real generalization performance is worse when SYNTHIA is used as the source domain as opposed to GTAV – for instance, ISW (Choi et al., 2021) achieves an average mIoU of 31.07% (SYNTHIA) as opposed to 35.58% (GTAV). SYNTHIA has significantly fewer images compared to GTAV (9.4k vs 25k), which likely contributes to relatively worse generalization performance.

**PASTA and Base Augmentations.** As stated in Sec. A.1, PASTA is applied with some consistent color and positional augmentations. To understand if PASTA alone leads to any improvements, we trained a baseline DeepLabv3+ model (R-50) without these augmentations. We find that from the settings reported in Table. 1a, Row 4 of the main paper plus training at a resolution of $1024 \times 560$, average performance on real datasets drops from (1) 41.90% to 40.37% mIoU when the photometric augmentations (color jitter and gaussian blur) are removed and (2) further drops to 40.25% mIoU when both positional and photometric augmentations are removed. Therefore, applying PASTA without any base augmentations leads to performance that is still significantly above a vanilla baseline (40.25% vs 26.99% mIoU).

## A.3 PASTA ANALYSIS

In this section, we provide more discussions surrounding different aspects of PASTA.

**Functional form of PASTA.** We now discuss the specific functional form of the perturbations introduced by PASTA as per Eqns. 4 and 5 in the main paper. Eqn. 5 is constructed to ensure we have

a multiplicative jitter interaction. Unlike the multiplicative interaction, an additive jitter interaction in the Fourier space where the perturbation is sampled from a gaussian distribution can be expressed as gaussian jitter perturbation in the RGB space – easier to attain by adding gaussian noise to the images. For Eqn. 5, we first considered uniform perturbations ($\alpha = 0$) independent of the spatial frequencies. As a natural extension based on the observations made in Section 3.2 of the main paper, we considered a linear dependence on spatial frequencies – hence, the inclusion of $\alpha$. The inclusion of $k$ decides how much attention we pay to the high-frequency bands compared to the low frequency ones. For fixed $\alpha$, since the frequency dependent term being exponentiated in Eqn. 5 is normalized $(2\sqrt{\frac{m^2+n^2}{H^2+W^2}} \in [0, 1])$, increasing $k$ perturbs the lower frequency components less.

**PASTA and Amplitude Mixup (AM).** We also compare PASTA with the Amplitude Mixup (AM) technique proposed in (Xu et al., 2021). Amplitude Mixup involves perturbing the amplitude spectrum of the image of concern by performing a convex combination with the amplitude spectrum of another "mixup" image drawn from the same source data. For a DeepLabv3+ architecture with a R-50 backbone, we find that when trained on GTAV (single run trained at a resolution of $1024 \times 560$), AM achieves an mIoU of 39.70% (vs 41.90% for PASTA) on the real datasets. We further observe that for AM, performance isn't altered even if we reduce the set of mixup images to a very small set (10 randomly selected images from GTAV) instead of sampling one for every source image. We observe that this variant of AM also ends up achieving an mIoU of 39.70% (single run). Overall, we observe that while AM tends to underperform compared to PASTA, the pipeline involved in AM – sampling "mixup" images – ends up being an overkill.

**PASTA and Amplitude Jitter (AJ).** We also compare PASTA with the Amplitude Jitter (AJ) technique considered in (Xu et al., 2021). Amplitude Jitter (AJ) perturbs the amplitude spectrum with a single jitter value $\epsilon$ for every spatial frequency and channel. $\epsilon$ is a multiplicative factor sampled from a gaussian distribution centered at 1. We use AJ with $\epsilon \sim \mathcal{N}(1, 0.5)$ for our experiments with DeepLabv3+ (R-50). We find that AJ ends up achieving an mIoU of 30.70% (single run trained at a resolution of $1024 \times 560$) on the real datasets on average, which is significantly lower than PASTA. Qualitatively, we observe that multiplying the entire amplitude spectrum by a constant value results in a uniform change of brightness in the augmented image. If we sample $\epsilon$ separately per-channel, we find that the performance of this AJ variant improves to an mIoU of 32.83% (single run trained at a resolution of $1024 \times 560$). Here, we observe that multiplying the amplitude spectrum channel-wise by three distinct constants leads to one of image channels dominating over the others, resulting in a red-ish, green-ish or blue-ish hue in the augmented images. Finally, if we consider PASTA ($\alpha = 0, \beta = 0.5$), which is a more fine-grained setting where $\epsilon$ is sampled per-spatial frequency and per-channel, we find that generalization performance improves significantly to 39.19% mIoU. Therefore, we observe that approaches along the lines of AJ for generalization lead to better results if implemented in a fine-grained manner as PASTA.

**Constrained PASTA ($\alpha = 0$).** Note that sampling an $\epsilon$ per spatial frequency creates a jitter image per-channel, which is then multiplied in an elementwise manner to the original amplitude to perturb the same. Similar to the Amplitude Mixup (AM) set of experiments, we also consider a setting where we sample these jitter images from a fixed set, instead of sampling a new one for every source image. We create these jitter image sets (each jitter image randomly sampled) of sizes 1, 2, 5 and 10 and observe that for DeepLabv3+ (R-50) (single runs trained at a resolution of $1024 \times 560$) we achieve mIoUs of 39.81%, 40.33%, 38.78%, and 39.16% respectively on the real datasets. Therefore, even if we severely restrict the source of randomness / variations in $\epsilon$ for PASTA ($\alpha = 0$), we don't observe a significant loss in generalization performance.

**Comparison with Fourier Domain Adaptation (FDA) (Yang & Soatto, 2020).** FDA is a recent approach for syn-to-real domain adaptation and naturally requires access to unlabeled target data. In FDA, low frequency bands of the amplitude spectra of source images are replaced with those of target – essentially mimicking a cheap style transfer operation. Since we do not assume access to target data in our experimental settings, a direct comparison is not possible. Instead, we consider a proxy task where we intend to generalize to real datasets (Cityscapes, BDD-100K, Mapillary) by assuming additional access to 6 real world street view images under different weather conditions (for style transfer) – sunny day, rainy day, cloudy day, etc. – in addition to synthetic images from GTAV. We find that FDA in this setting achieves an mIoU of 33.04% (vs 26.99% for Baseline and 41.90% for Baseline + PASTA ($\alpha = 3, k = 2$)) for DeepLabv3+ with an R-50 backbone trained at a resolution of $1024 \times 560$.

---

**Algorithm 1** Pseudocode of PASTA in a PyTorch-like style.

---

```
# image: CxHxW input image for PASTA, pixel values are 0 to 1
# a, b, k: alpha, beta, and k PASTA hyperparameters

fft = fft(image, dim=[1, 2]) # get 2D fft for each channel
amp = abs(fft) # get amplitude of image
phase = angle(fft) # get phase of image

C,H,W = image.shape # get channel, height, and width of image

m = range(-H/2, H/2) # range for m
n = range(-W/2, W/2) # range for n

MM,NN = meshgrid(n,m) # HxW grid of m values and n values
dist_grid = sqrt(NN**2 + MM**2) # HxW grid where each entry is distance from center pixel
norm_dist_grid = dist_grid / sqrt(H**2 + W**2) # normalize distance grid by diagonal length

sigma = ((2 * a * norm_dist_grid) ** k) + b # HxW grid of standard deviations
sigma = tile(sigma, (C, 1, 1)) # CxHxW grid of standard deviations

# CxHxW array where an entry at [m,n] is sampled from N(1, sigma[m,n])
epsilon = normal(mean=1, std=sigma)

pasta_amp = amp * epsilon # PASTA-perturbed amplitude

# reconstruct fft using perturbed amplitude and original phase
pasta_fft = pasta_amp * exp(1j * phase)
pasta_image = ifft(pasta_fft, dim=[1, 2]) # take 2D inverse fft for each channel
pasta_image = real(pasta_image) # keep real values from inverse fft
pasta_image = clip(pasta_image, 0, 1) # constrain pixel values from 0 to 1
return pasta_image
```

---

**Overall sensitivity to $\alpha, k$.** Careful selection of $\alpha, k$ is more important compared to $\beta$. Similar to our observations on VisDA-C (Section 5.2 main paper), we find that restricting the $k \in [1, 4]$ offers stable improvements in semantic segmentation GTAV→Real for a DeepLabv3+ model (RN-50) backbone trained at a resolution of 1024x560 (see Table. 8). For $k \in [1, 4]$, to obtain stable improvements for a vanilla baseline (not designed specifically for syn-to-real generalization), restricting $\alpha \in (0, 4]$ offers stable generalization improvements. Exceeding these ranges leads to severe augmentations that obfuscate task-relevant semantic information.

Table 8: **Ablating $k$ for GTAV as synthetic source.** We apply PASTA with varying values of $k$ to a DeepLabv3+ (RN-50) semantic segmentation trained at a source input resolution of $1024 \times 560$ for the GTAV→Real (Cityscapes, BDD-100K, Mapillary) task.

| Method | mIoU (Real) |
|---|---|
| 1 Baseline | 26.99 |
| **Ablating $k$ for ($\alpha = 3$)** | |
| 2 + PASTA ($k = 1$) | 41.50 |
| 3 + PASTA ($k = 2$) | **41.90** |
| 4 + PASTA ($k = 4$) | 41.09 |
| 5 + PASTA ($k = 6$) | 30.90 |

### A.4 AMPLITUDE ANALYSIS

PASTA is motivated by the empirical observation that synthetic images have less variance in their high frequency components compared to real images. In this section, we first show how this observation is widespread across a set of syn-to-real shifts over fine-grained frequency band discretizations and then demonstrate how PASTA helps counter this discrepancy.

**Fine-grained Band Discretization.** For Fig. 2 (b) in the main paper, the low, mid and high frequency bands are chosen such that the first band is $1/3$ the height of the image (includes all spatial frequencies till $1/3$rd of the image height), second band is up to $2/3$ the height of the image excluding band 1 frequencies, and the third band considers all the remaining frequencies. We begin by splitting the amplitude spectrum into 3, 5, 7, and 9 frequency bands in the manner described above, and analyze the diversity of these frequency bands across multiple datasets. Across 7 domain shifts (see Fig. 5 and 6) – {GTAV, SYNTHIA} → {Cityscapes, BDD-100K, Mapillary}, and VisDA-C Syn→Real, we find that (1) for every dataset (whether synthetic or real), diversity decreases as we head towards higher frequency bands and (2) synthetic images exhibit less diversity in high-frequency bands at all the levels of granularity we consider.

**Increase in amplitude variations post-PASTA.** Next, we observe how PASTA effects the diversity of the amplitude spectrums on GTAV and VisDA-C. Similar to above, we split the amplitude spectrum into 3, 5, 7, and 9 frequency bands, and we analyze the diversity of these frequency bands before and after applying PASTA to images (see Fig. 7 and 8). For synthetic images from GTAV, when

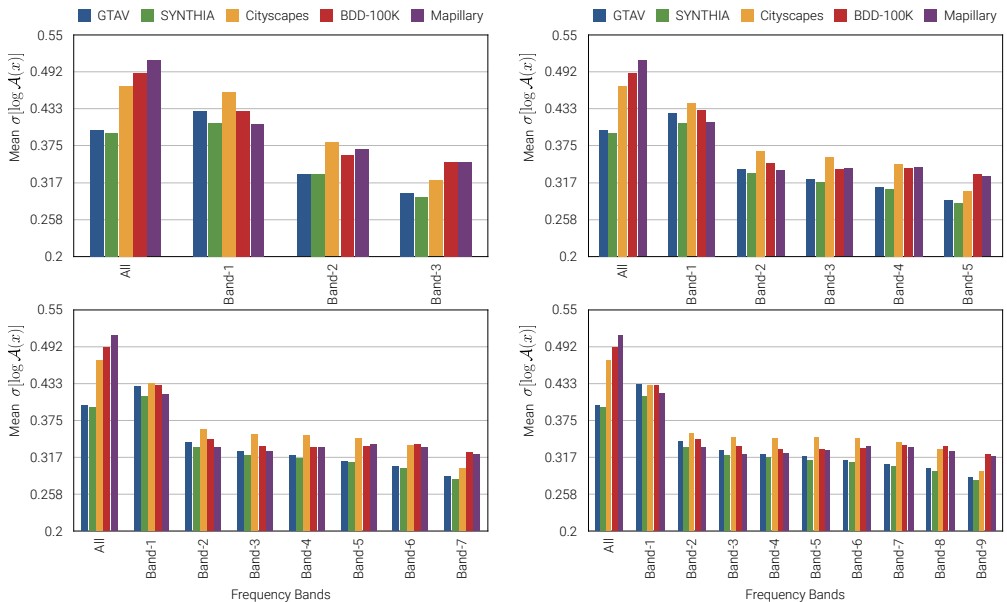

Figure 5: **Variations in amplitude values across fine-grained frequency bands (GTAV→Real and SYNTHIA→Real).** Across domain shifts GTAV→Real and SYNTHIA→Real, and four settings corresponding to fine-grained frequency bands (3, 5, 7 and 9 bands; increasing in frequency from Band-1 to Band-$n$), we find that synthetic images have less variance in high-frequency components of the amplitude spectrum compared to real images.

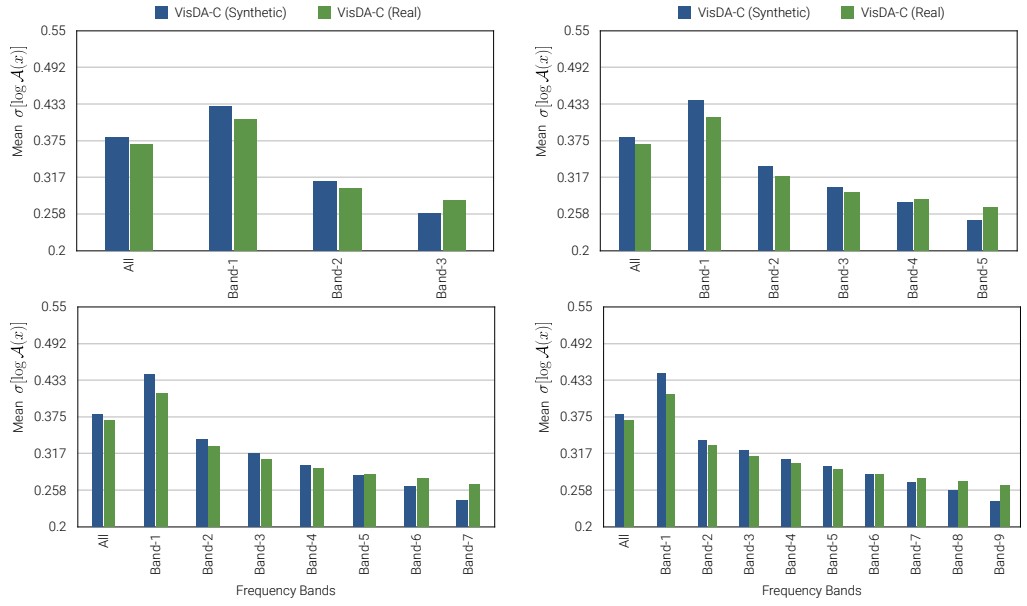

Figure 6: **Variations in amplitude values across fine-grained frequency bands (VisDA-C Synthetic→Real).** For the VisDA-C Synthetic→Real domain shift, and four settings corresponding to fine-grained frequency bands (3, 5, 7 and 9 bands; increasing in frequency from Band-1 to Band-$n$), we find that synthetic images have less variance in high-frequency components of the amplitude spectrum compared to real images.

PASTA ($\alpha = 3, k = 2$) is applied, we observe that the standard deviation of amplitude spectrums increases from $0.4$ to $0.497$, $0.33$ to $0.51$ and $0.3$ to $0.52$ for the low, mid and high frequency bands respectively. As expected, we observe maximum increase for the high-frequency bands.

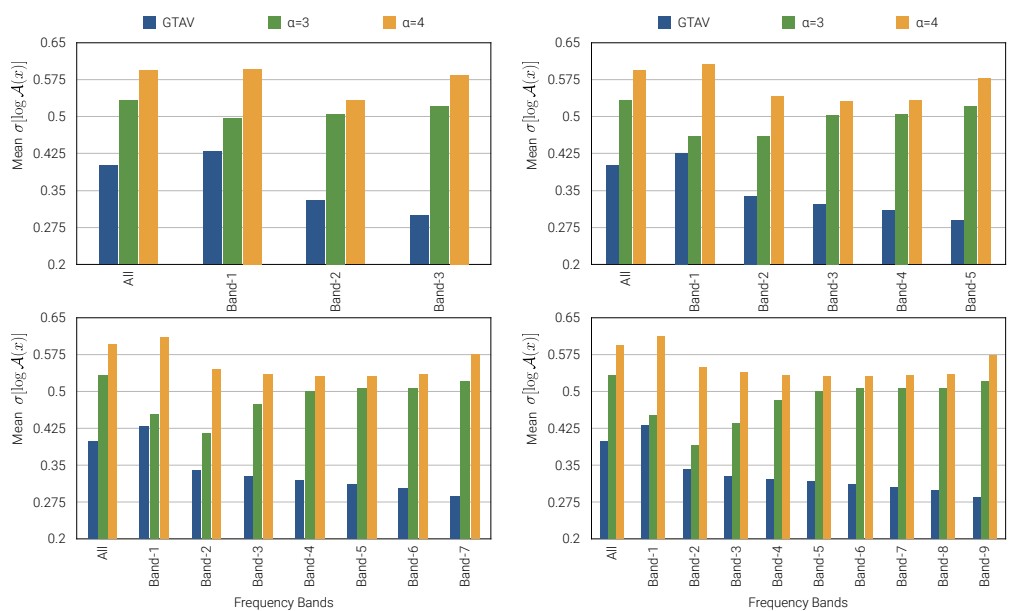

Figure 7: **Variations in amplitude values across fine-grained frequency bands for synthetic images post-PASTA (GTAV).** For GTAV, we find that applying PASTA increases variations in amplitude values across different frequency bands. Four plots correspond to fine-grained frequency bands (3, 5, 7 and 9 bands; increasing in frequency from Band-1 to Band-$n$). We find the maximum amount of increase for the highest frequency bands across different granularity levels.

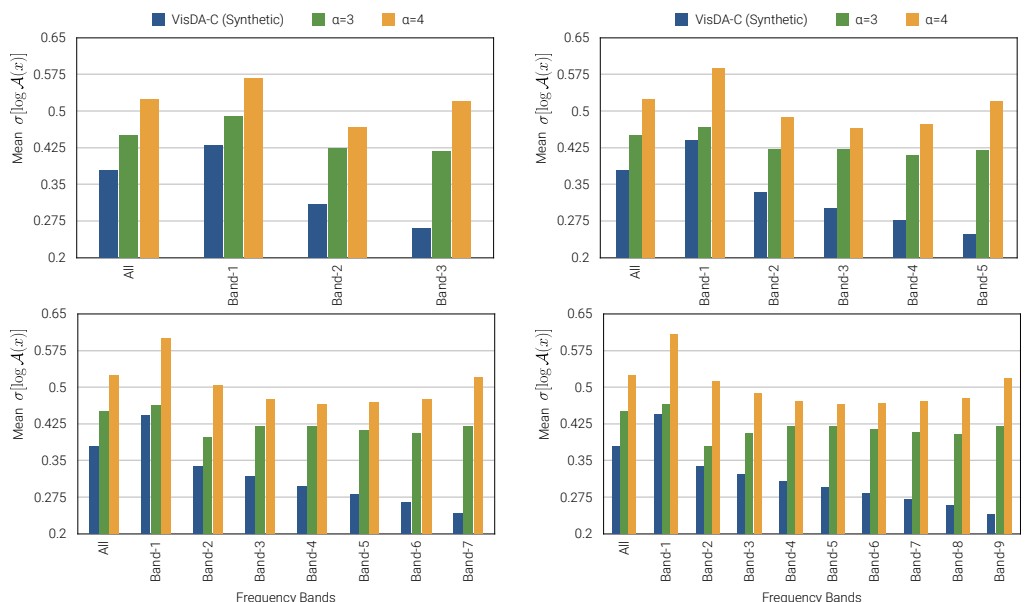

Figure 8: **Variations in amplitude values across fine-grained frequency bands for synthetic images post-PASTA (VisDA-C).** For VisDA-C (Synthetic), we find that applying PASTA increases variations in amplitude values across different frequency bands. Four plots correspond to fine-grained frequency bands (3, 5, 7 and 9 bands; increasing in frequency from Band-1 to Band-$n$). We find the maximum amount of increase for the highest frequency bands across different granularity levels.

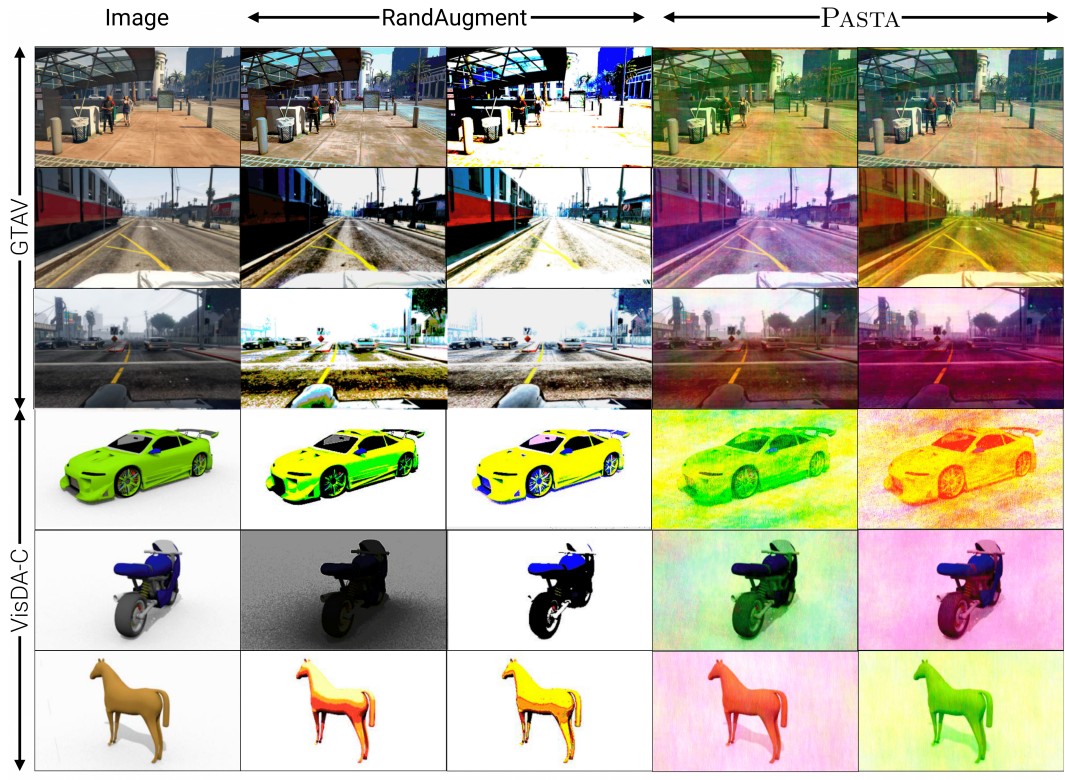

Figure 9: **PASTA augmentation samples.** Examples of images from different synthetic datasets when augmented using PASTA and RandAugment (Cubuk et al., 2020). Rows 1-3 include examples from GTAV and rows 4-6 from VisDA-C.

## A.5 QUALITATIVE EXAMPLES

**PASTA Augmentation Samples.** Fig. 9 includes more examples of images from synthetic datasets (from GTAV and VisDA-C), when RandAugment and PASTA are applied.

**Semantic Segmentation Predictions.** We include qualitative examples of semantic segmentation predictions on Cityscapes made by Baseline, IBN-Net and ISW (DeepLabv3+, ResNet-50) trained on GTAV (no downsampling, corresponding to Table. 1a in the main paper) Fig. 10, 12 and 14 respectively when different augmentations are applied (RandAugment and PASTA). The Cityscapes images we show predictions on were selected randomly. We include RandAugment predictions only for the Baseline. To get a better sense of the kind of mistakes made by different approaches, we also include the difference between the predictions and ground truth segmentation masks in Fig. 11, 13 and 15 (ordered accordingly for easy reference). The difference images show the predicted classes only for pixels where the prediction differs from the ground truth.

## A.6 ASSETS LICENSES

The assets used in this work can be grouped into three categories – Datasets, Code Repositories and Dependencies. We include the licenses of each of these assets below.

**Datasets.** We used the following publicly available datasets in this work – GTAV (Richter et al., 2016), Cityscapes (Cordts et al., 2016), BDD-100K (Yu et al., 2020), Mapillary (Neuhold et al., 2017), and VisDA-C (Peng et al., 2017). For GTAV, the codebase used to extract data from the original GTAV game is distributed under the MIT license.[5] The license agreement for the Cityscapes dataset dictates that the dataset is made freely available to academic and non-academic entities for non-commercial purposes such as academic research, teaching, scientific publications, or personal

---

[5]https://bitbucket.org/visinf/projects-2016-playing-for-data/src/master/

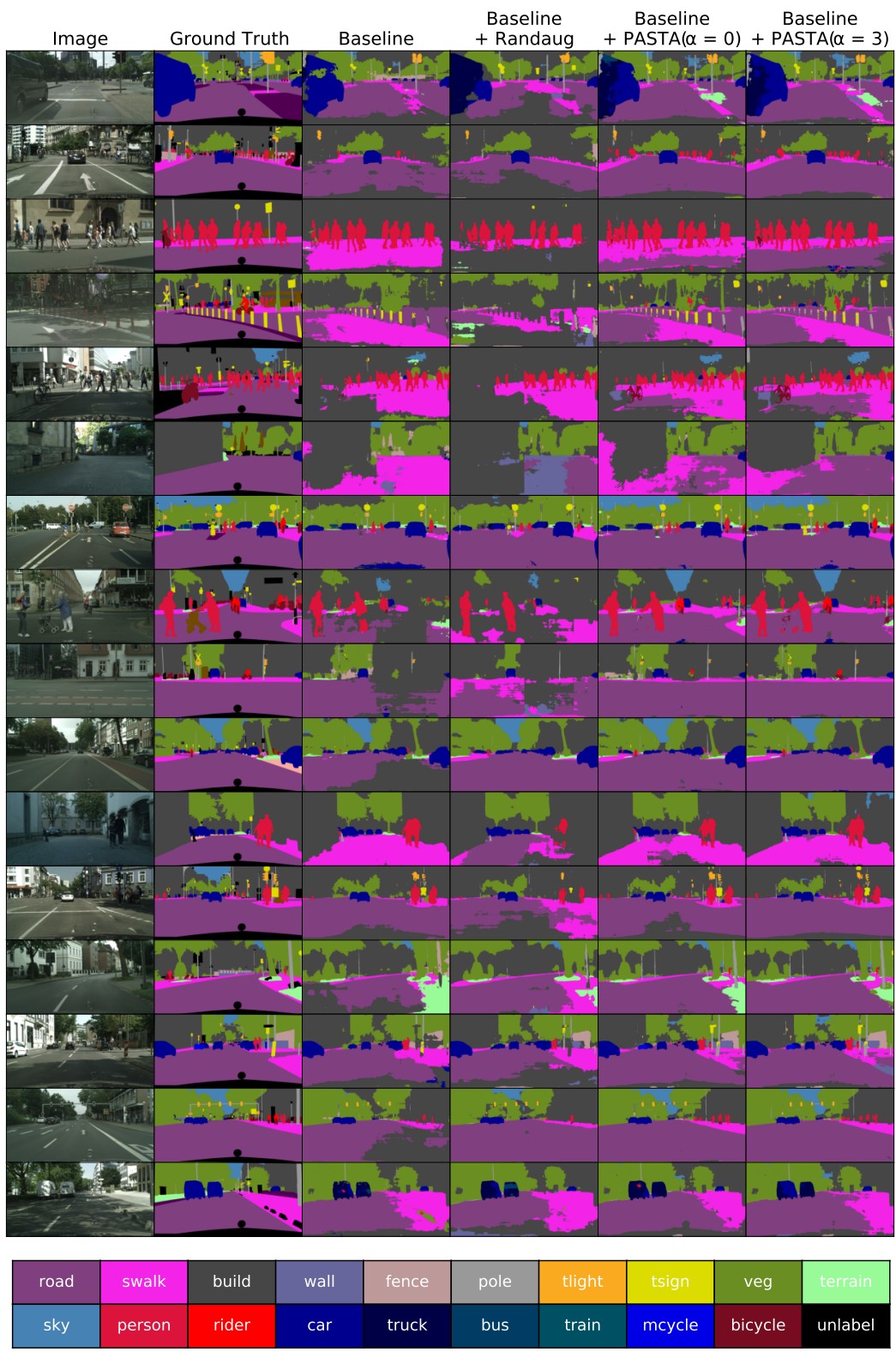

Figure 10: **GTAV→Cityscapes Baseline segmentation predictions.** Qualitative predictions made on randomly selected Cityscapes validation images by a Baseline DeepLabv3+ model (R-50 backbone) trained on GTAV synthetic images. The first two columns indicate the original image and the associated ground truth and rest indicate the considered approaches.

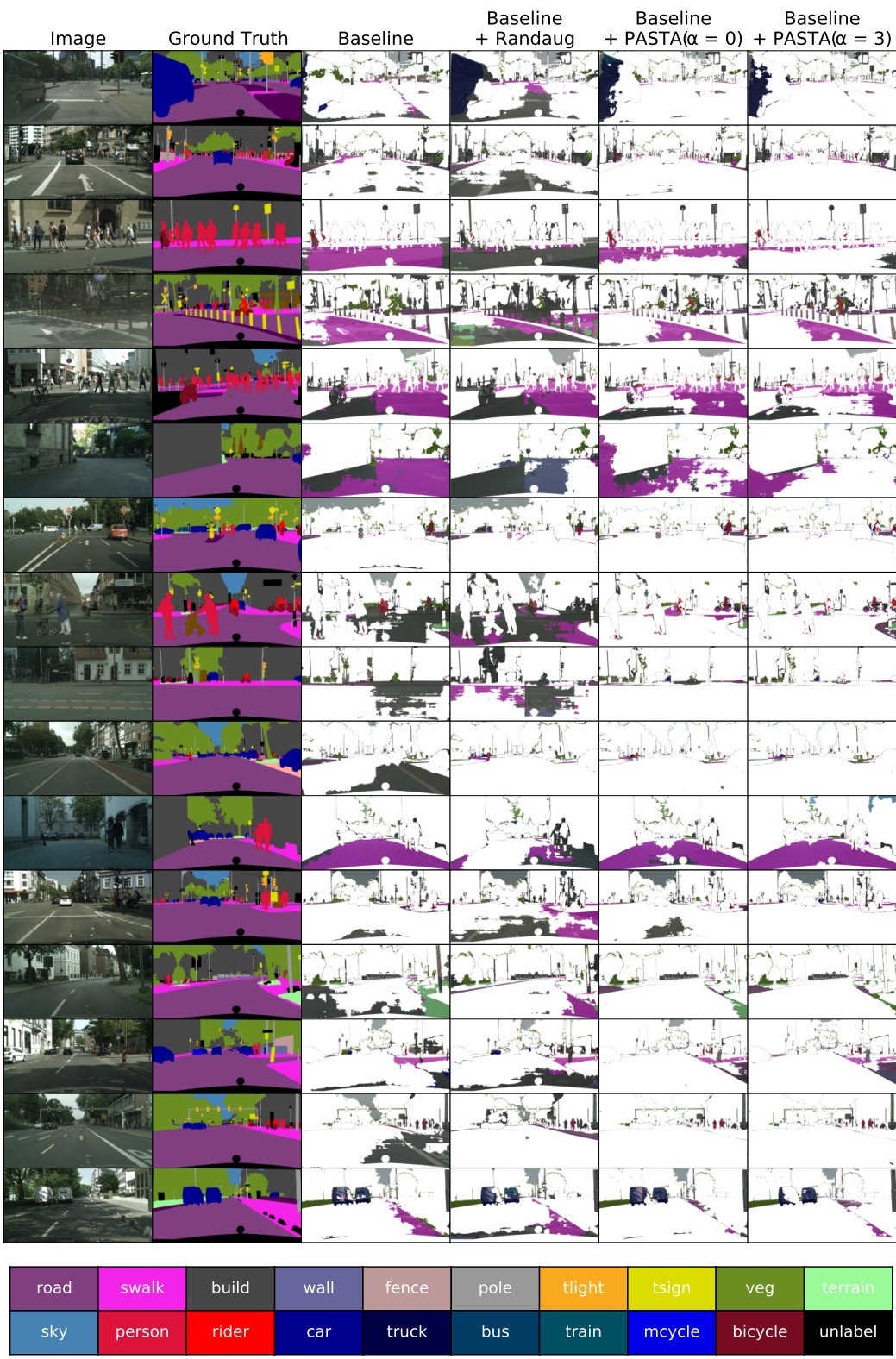

Figure 11: **GTAV→Cityscapes Baseline segmentation prediction diffs.** Differences between prediction and ground truth for predictions made on randomly selected Cityscapes validation images by a Baseline DeepLabv3+ model (R-50 backbone) trained on GTAV synthetic images. The first two columns indicate the original image and the associated ground truth and rest indicate the considered approaches.

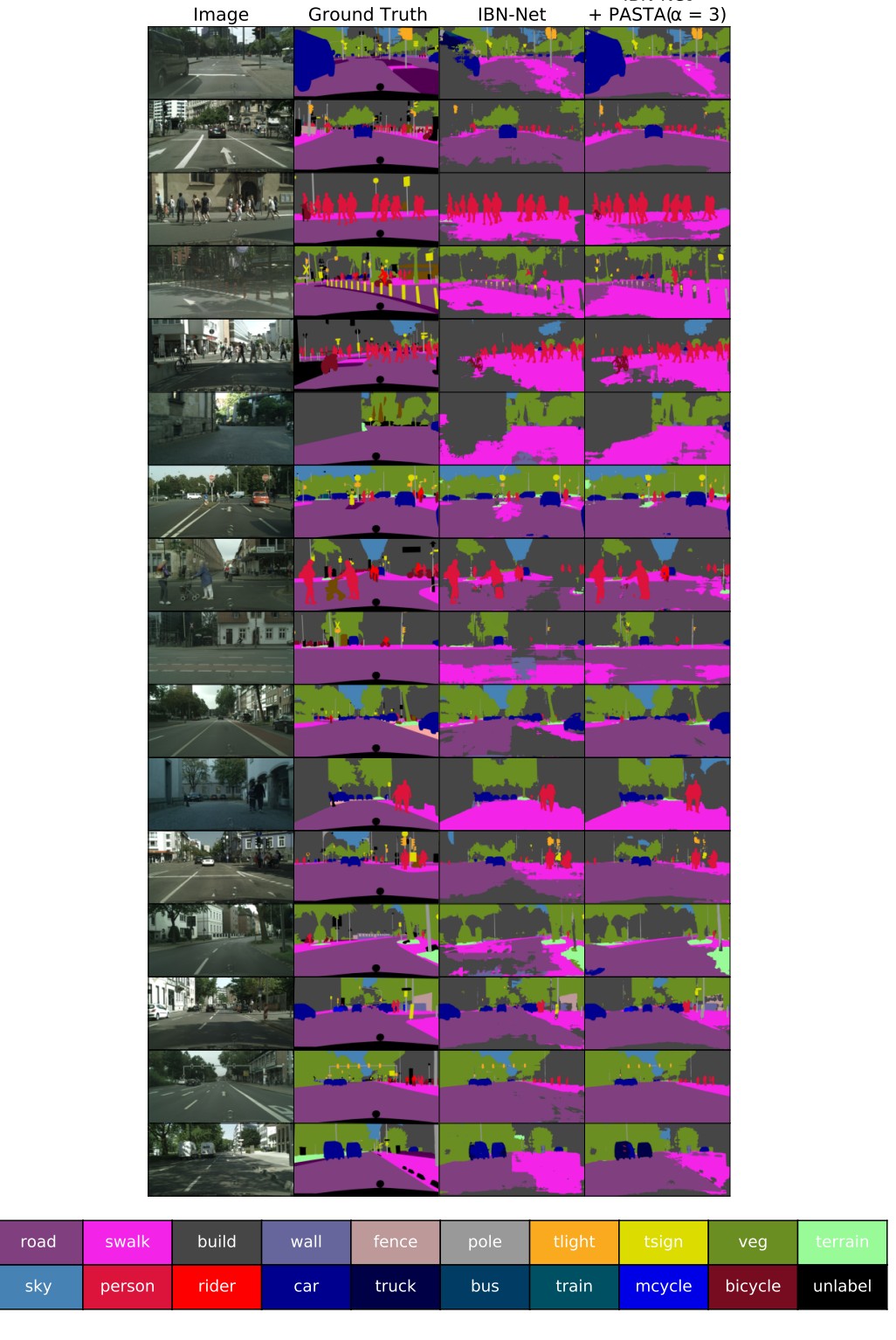

| road | swalk | build | wall | fence | pole | tlight | tsign | veg | terrain |
|---|---|---|---|---|---|---|---|---|---|
| sky | person | rider | car | truck | bus | train | mcycle | bicycle | unlabel |

Figure 12: **GTAV→Cityscapes IBN-Net (Pan et al., 2018) segmentation predictions.** Qualitative predictions made on randomly selected Cityscapes validation images by IBN-Net (DeepLabv3+ model with R-50 backbone) trained on GTAV synthetic images. The first two columns indicate the original image and the associated ground truth and rest indicate the considered approaches.

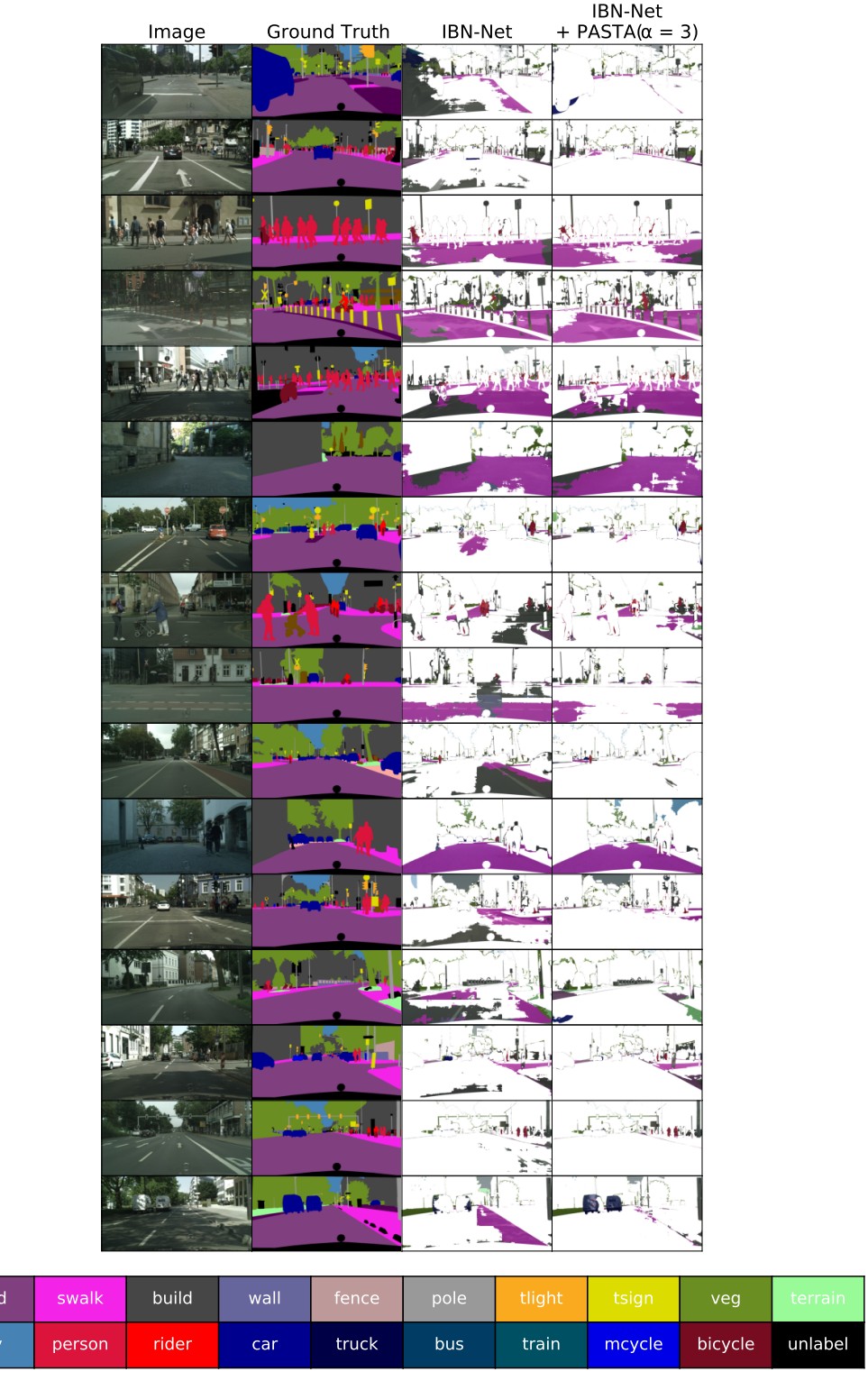

Figure 13: **GTAV→Cityscapes IBN-Net (Pan et al., 2018) segmentation prediction diffs.** Differences between prediction and ground truth for predictions made on randomly selected Cityscapes validation images by IBN-Net (DeepLabv3+ model with R-50 backbone) trained on GTAV synthetic images. The first two columns indicate the original image and the associated ground truth and rest indicate the considered approaches.

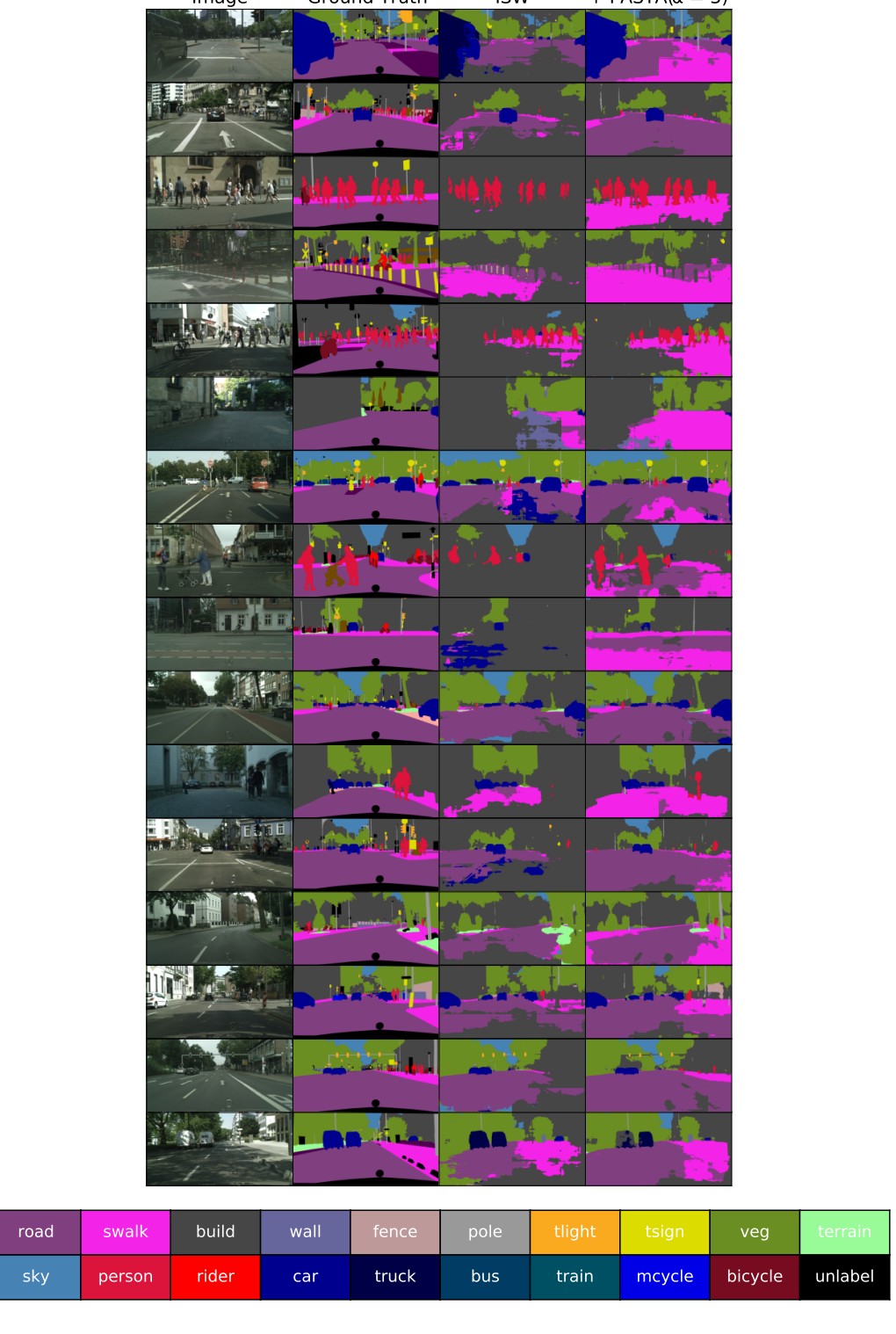

Figure 14: **GTAV→Cityscapes ISW (Choi et al., 2021) segmentation predictions.** Qualitative predictions made on randomly selected Cityscapes validation images by ISW (DeepLabv3+ model with R-50 backbone) trained on GTAV synthetic images. The first two columns indicate the original image and the associated ground truth and rest indicate the considered approaches.

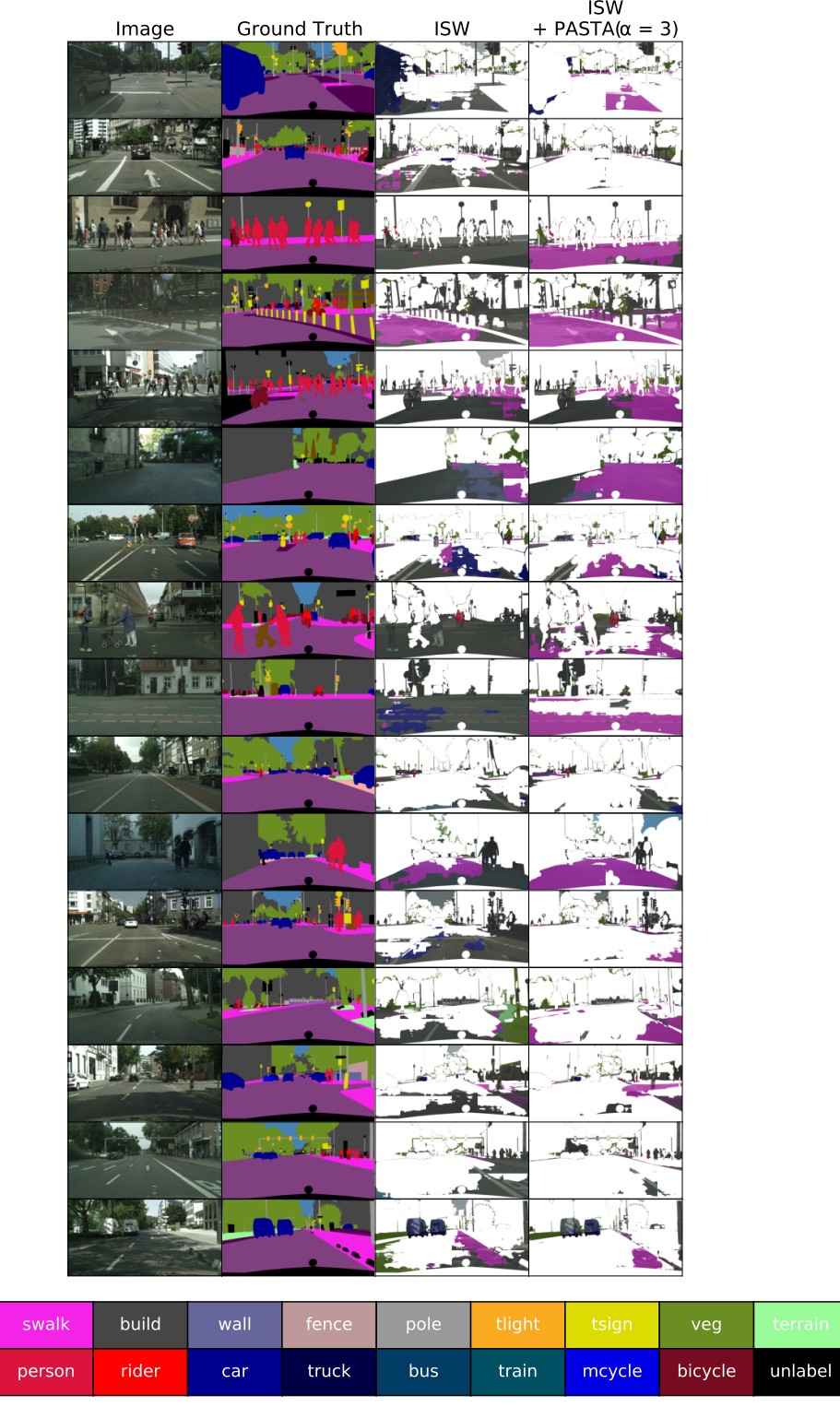

| road | swalk | build | wall | fence | pole | tlight | tsign | veg | terrain |
|------|-------|-------|------|-------|------|--------|-------|-----|---------|
| sky | person | rider | car | truck | bus | train | mcycle | bicycle | unlabel |

Figure 15: **GTAV→Cityscapes ISW (Choi et al., 2021) segmentation prediction diffs.** Differences between prediction and ground truth for predictions made on randomly selected Cityscapes validation images by ISW (DeepLabv3+ model with R-50 backbone) trained on GTAV synthetic images. The first two columns indicate the original image and the associated ground truth and rest indicate the considered approaches.

experimentation and that permission to use the data is granted under certain conditions.[6] BDD-100K is distributed under the BSD-3-Clause license.[7] Mapillary images are shared under a CC-BY-SA license, which in short means that anyone can look at and distribute the images, and even modify them a bit, as long as they give attribution.[8] The VisDA-C development kit on github does not have a license associated with it, but it does include a Terms of Use, which primarily states that the dataset must be used for non-commercial and educational purposes only.[9]

**Code Repositories.** For our experiments, apart from code that we wrote ourselves, we build on top of three existing public repositories – RobustNet[10], MMDetection[11] and CSG[12]. RobustNet is distributed under the BSD-3-Clause license. MMDetection is distributed under Apache License 2.0[13]. CSG, released by NVIDIA, is released under a NVIDIA-specific license.[14]

**Dependencies.** We use Pytorch Paszke et al. (2019) as the deep-learning framework for all our experiments. Pytorch, released by Facebook, is distributed under a Facebook-specific license.[15]

---

[6] https://www.cityscapes-dataset.com/license/

[7] https://github.com/bdd100k/bdd100k/blob/master/LICENSE

[8] https://help.mapillary.com/hc/en-us/articles/115001770409-Licenses

[9] https://github.com/VisionLearningGroup/taskcv-2017-public/tree/master/classification

[10] https://github.com/shachoi/RobustNet

[11] https://github.com/open-mmlab/mmdetection

[12] https://github.com/NVlabs/CSG

[13] https://github.com/open-mmlab/mmdetection/blob/master/LICENSE

[14] https://github.com/NVlabs/CSG/blob/main/LICENSE.md

[15] https://github.com/pytorch/pytorch/blob/master/LICENSE

