# OpenReview forum: "Proportional Amplitude Spectrum Training Augmentation for Synthetic-to-Real Domain Generalization"
_ICLR.cc/2023/Conference — Submitted to ICLR 2023_

### Official Review · Reviewer_tiZd · 2022-10-22

**Confidence:** 4
**Correctness:** 2
**Technical Novelty And Significance:** 2
**Empirical Novelty And Significance:** 2
**Recommendation:** 5

**Clarity, Quality, Novelty And Reproducibility:**

The idea is rather easy to follow. The experiment part requires major revision, see the comments above.

**Strength And Weaknesses:**

Strength:
1. Simple approach but rather effective compared with some more complex methods.
2. The effectiveness is showed across different tasks, e.g., semantic segmentation, object detection.

Weakness:
1. Frequency domain augmentation is an existing augmentation strategy, see listed papers below. It is also reviewed in the related work of this paper. However, there generally miss direct comparison with data augmentation schemes, especially those which are also based on spectrum perturbation. In the paper, the authors mainly compared with one and reported the number in the text for a single task. This is far from sufficient.
2. The perturbation design is rather hand-crafted. It's also sensitive to hyperparameters such as alpha, k, and beta(also mentioned in the limitations). Therefore, without comprehensive comparisons with prior work on frequency domain augmentation, PASTA is not a convincing new solution.
3. The scheme is more effective in SemSeg, whereas the performance in object detection is not convincing.
4. There are multiple questionable places in the experiment part:
- In Tab. 1-d), it is strange why a larger backbone would cause worse performance.
- For SemSeg in Tab 1, it is also unclear why the authors did not choose the best performing method WildNet as the baseline and apply PASTA with it for a better generalization performance.
- The in-distribution performance is not provided in Table 1.
- From Table 2, PASTA (obviously)  hurts the ID performance. Is this always the case and any explanation for this?

Papers related to frequency domain augmentation:
- Amplitude-Phase Recombination: Rethinking Robustness of Convolutional Neural Networks in Frequency Domain
- Frequency Spectrum Augmentation Consistency for Domain Adaptive Object Detection
- Domain Generalization via Frequency-domain-based Feature Disentanglement and Interaction
- Amplitude Spectrum Transformation for Open Compound Domain Adaptive Semantic Segmentation
- A Fourier-based Framework for Domain Generalization



**Summary Of The Paper:**

This paper proposed a data augmentation method PASTA for syn2real generalization. Based on the finding that real images exhibit larger high frequency variances than synthetic ones, PASTA perturbs the high frequency bands of the amplitude spectrum of the synthetic images. In doing so, the high frequency variance is larger while the semantic content is preserved. This augmentation scheme is applicable to different tasks.

**Summary Of The Review:**

In general, the empirical evidence is not sufficient, whereas the algorithm itself is rather hand-crafted.

---

> ### Author Response · Authors · 2022-11-12
> **Response to Reviewer tiZd  (1/3)**
>
> We thank the reviewer for their feedback and respond to specific concerns below.
>
> #### **Comparison to Other Frequency Augmentation Methods**
> >However, there generally miss direct comparison with data augmentation schemes, especially those which are also based on spectrum perturbation. In the paper, the authors mainly compared with one and reported the number in the text for a single task. This is far from sufficient.
>
> We would like to respectfully point out that we **do** compare against recent frequency-based methods in the main paper (and the appendix). We want to highlight that **all** prior works operate in different settings which assumed access to extra data (either additional source data or access to target data).
>
> In addition, our related work section cites references to multiple existing works that leverage the frequency domain for data augmentation. We thank the reviewer for the additional pointers and provide a discussion or comparison to the works mentioned.
>
> PASTA is designed to improve syn-to-real domain generalization. Thus, it does not assume access to any real data during training and instead, the same model is evaluated out of the box on multiple different real-world datasets. Many of the works the reviewer mentioned including [C, D] are designed for domain adaptation, *i.e.,* assume access to and evaluation on a single real target domain at a time.
>
> * **Comparison to FSDR [A]:** We provide comparisons to FSDR [A] in Table 1(b). Note that FSDR relies on extra data ($\sim2.8$x more images) from both GTAV (explicitly) and SYNTHIA (implicitly) as synthetic sources. Leveraging this extra data results in a final performance of 43.13 with RN-101, DeepLabv3+. In comparison, PASTA augmentation combined with standard ERM training and **no extra data** yields 42.01 mIoU in this setting. While these numbers cannot be directly compared due to FSDR using extra data, it is a signal that PASTA is competitive with FSDR.
>
> * **Comparison to [FDA], [APR], and [FACT]**: We provide a comparison to [FDA] in Appendix A.3 and a comparison to [FACT] in Section 5.1 of the main paper. We provide new comparisons to [APR] following the reviewer's suggestion and present overall results in the table below.
>     * **Setting**: We evaluate on the setting of semantic segmentation GTAV$\rightarrow${Cityscapes \(C\), BDD-100K (B), Mapillary (M)}, DeepLabv3+ model (RN-50) backbone trained on GTAV at a resolution of 1024x560
>     * **FDA**: FDA is a domain adaptation method and therefore requires access to target data. The method replaces low-frequency bands of source amplitudes with those of target images. Since this paper focuses on out-of-the-box generalization, meaning performance before target data is observed, FDA cannot be directly applied.  However, in an attempt to compare against prior work, we computed the performance for FDA if given a few real images (denoted FDA*) and found the approach to significantly underperform PASTA (see Appendix A.3).
>     * **APR**: This method was designed to improve robustness against natural corruption. APR replaces the amplitude spectrum of an image with the amplitude spectrum from an augmented view (APR-S) or different images (APR-P). It also underperforms PASTA.
>     * **FACT**: We compare the frequency domain augmentation schemes adopted in FACT -- Amplitude Mixup (AM) and Amplitude Jitter (AJ) -- with PASTA. We find that (1) PASTA outperforms FACT-Aug (AJ) and (2) PASTA outperforms FACT-Aug (AM) without the overhead of sampling a replacement amplitude spectrum.
>
> |  | Method  | Avg mIoU {C, B, M}  |
> |---|---| ---|
> | 1|Baseline | 26.99  |
> | 2|FDA* | 33.04 |
> | 2|Baseline + APR-P | 37.52  |
> | 3|Baseline + FACT-Aug (AJ) | 30.70  |
> | 4|Baseline + FACT-Aug (AM) | 39.70  |
> | 5|Baseline + PASTA | **41.90**  |
>
> (Models trained at an input resolution of 1024x560)
>
> * **Comparison to [FFDI]**: FFDI is contemporaneous work (following ICLR 2023 guidelines [E]) and was also designed for multi-source DG settings. Unlike PASTA, FFDI applies unstructured perturbations to both the amplitude and phase spectra of images.
>
> [A] FSDR: Frequency space domain randomization for domain generalization (CVPR 2021)
>
> [B] FDA: Fourier domain adaptation for semantic segmentation (CVPR 2020)
>
> [C] Amplitude Spectrum Transformation for Open Compound Domain Adaptive Semantic Segmentation (AAAI 2022)
>
> [D] Frequency Spectrum Augmentation Consistency for Domain Adaptive Object Detection (arXiv 2021)
>
> [**APR**] Amplitude-Phase Recombination: Rethinking Robustness of Convolutional Neural Networks in Frequency Domain (ICCV 2021)
>
> [**FACT**] A Fourier-based Framework for Domain Generalization (CVPR 2021) (Code: https://github.com/MediaBrain-SJTU/FACT)
>
> [**FFDI**] Domain Generalization via Frequency-domain-based Feature Disentanglement and Interaction (ACM Multimedia 2022) (Code: https://github.com/PRIS-CV/FFDI)
>
> [E] - https://iclr.cc/Conferences/2023/ReviewerGuide

---

> > ### Author Response · Authors · 2022-11-12
> > **Response to Reviewer tiZd (2/3)**
> >
> > **PASTA Algorithm Design**
> > >The perturbation design is rather hand-crafted.
> >
> > Thanks for bringing this up. For ease of reference, we provide Equations (4) and (5) from our main paper which specifies our amplitude perturbation approach below.
> >
> > $g_{\Lambda}(\mathcal{A}(x))[m,n] = \epsilon[m,n] \mathcal{A}(x)[m,n]$ **(Eqn 4)**
> >     where $\epsilon[m,n]\sim\mathcal{N}(1,\sigma[m,n])\mbox{ and }\sigma[m,n] = \left(2\alpha\sqrt{\frac{m^2 + n^2}{H^2 + W^2}}\right)^k + \beta \mbox{ and }\Lambda=\lbrace{\alpha, k, \beta\rbrace}$ **(Eqn 5)**
> >
> > We reiterate and attempt to clarify the goal of the three hyperparameters for this equation:
> > * $\beta$: additive term, provides a constant level of variance. For semantic segmentation and object detection, we set $\beta$ to $0.25$. For object recognition, $\beta \in \lbrace{0.5,0.875\rbrace}$.
> > * $\alpha$: a linear factor applied to the spatial frequencies. As frequencies get higher the variance increases linearly with $\alpha$. For semantic segmentation and object detection, we set $\alpha$ to $3$. For object recognition, $\alpha \in \lbrace{4,10\rbrace}$
> > * $k$: An exponential factor applied to the spatial frequencies. For a fixed $\alpha$, since the frequency term being exponentiated in Eqn 5 is normalized ($2\sqrt{\frac{m^2 + n^2}{H^2 + W^2}} \in [0, 1]$), increasing $k$ perturbs the lower frequency components less. $k$ is set to $2$ in our experiments.
> >
> > Our equation for computing amplitude variance that varies with the spatial frequencies was carefully designed to match the observation that **compared to real images, synthetic images have lower variance in the high-frequency components**. Thus, Eqn's 4 and 5 allow for augmented variance to be added to synthetic data that can expose a model to more high-frequency variations during training. Our careful design of PASTA (as illustrated in Eqn. 5) leads to considerable improvements in synthetic-to-real generalization performance across multiple shifts and tasks.
> >
> > >It's also sensitive to hyperparameters such as alpha, k, and beta(also mentioned in the limitations). Therefore, without comprehensive comparisons with prior work on frequency domain augmentation, PASTA is not a convincing new solution.
> >
> > * **Different optimal $\alpha,k,\beta$ across synthetic sources**: Synthetic images from GTAV and VisDA-C differ significantly - GTAV includes street view images whereas VisDA-C images are synthetic 2D renderings of 3D objects under different viewpoints and lighting [F] (see Figure 9, appendix). Compared to GTAV, VisDA-C images are relatively less diverse in high-frequency components. These differences contribute to the choice of optimal hyper-parameters.
> >
> > * **Overall sensitivity to $\alpha,k$**: Careful selection of $\alpha,k$ is more important compared to $\beta$. For our experiments, we find that restricting the $k\in[1,4]$ offers stable improvements. In addition to the observations in Section 5.2 for VisDA-C, we show the same in the following table for semantic segmentation GTAV$\rightarrow$Real, for DeepLabv3+ model (RN-50) backbone trained at a resolution of 1024x560. For $k\in[1,4]$, to obtain stable improvements for a vanilla baseline (not designed for syn-to-real generalization), restricting $\alpha\in(0,4]$ offers stable generalization improvements. Exceeding these ranges leads to severe augmentations that obfuscate task-relevant semantic information. We include this discussion in appendix A.3 (in red).
> >
> > | | Method |mean mIoU (Real)|
> > |---|---|---|
> > |1 | Baseline|	26.99 |
> > | | **Ablating $k$ for $(\alpha=3)$**|	 |
> > |2 | + PASTA ($k=1$)|	41.50 |
> > |3 | + PASTA ($k=2$)|	41.90 |
> > |4 | + PASTA ($k=4$)|	41.09 |
> > |5 | + PASTA ($k=6$)|	30.90 |
> >
> > (Models trained at an input resolution of 1024x560)
> >
> > [F] - Classification Track – http://ai.bu.edu/visda-2017/
> >
> > **Object Detection Results**
> >
> > >The scheme is more effective in SemSeg, whereas the performance in object detection is not convincing.
> >
> > We respectfully disagree. In addition to Semantic Segmentation, we observe strong improvements in Object Detection as well (outlined in the table below for Sim10K$\rightarrow$Real). Additionally, these improvements are obtained by just introducing PASTA as an augmentation in a plug-and-play fashion to a baseline model during training.
> >
> > |  | Method  | Synthetic  | Real  | mAP@50 (RN-50)  | mAP@50 (RN-101)  |
> > |---|---|---|---|---|---|
> > | 1|Baseline | $\checkmark$ |  |  39.4  | 43.3 |
> > | 2|+ PASTA  | $\checkmark$ |  |  56.3 (+16.9)  | 55.2 (+11.9) |
> > | 3| ILLUME* | $\checkmark$  | $\checkmark$  | | 53.1 (+9.8)  |
> >
> > * **PASTA significantly improves vanilla baseline**: For a vanilla baseline (*not designed specifically for syn-to-real generalization*), applying PASTA improves performance by 16.9 and 11.9 absolute mAP for RN-50 and RN-101 respectively.
> > * **PASTA outperforms SOTA domain adaptation method**: For both RN-50 and RN-101, PASTA outperforms, ILLUME*, a domain adaptation method -- *that assumes access to real (target) data at training time*.

---

> > > ### Author Response · Authors · 2022-11-12
> > > **Response to Reviewer tiZd (3/3)**
> > >
> > > **Experimental Concerns**
> > > >There are multiple questionable places in the experiment part:
> > >
> > > We address experimental concerns below.
> > >
> > > >For SemSeg in Tab 1, it is also unclear why the authors did not choose the best-performing method WildNet as the baseline and apply PASTA with it for better generalization performance.
> > >
> > > Since WildNet [G] relies on using **extra real-images**, additional modeling components and objectives for synthetic image stylization and applying PASTA to a vanilla baseline results in performance either better than or competitive with WildNet, we did not consider applying PASTA to the same.
> > >
> > > [G] - WildNet: Learning Domain Generalized Semantic Segmentation from the Wild
> > >
> > > **Reduced Performance on Synthetic test data**
> > > >From Table 2, PASTA (obviously) hurts the ID performance. Is this always the case and any explanation for this?
> > > >The in-distribution performance is not provided in Table 1.
> > >
> > > Yes. PASTA modestly reduces in-domain performance on **synthetic data** (see Table below for GTAV$\rightarrow$Synthetic, DeepLabv3+, RN-50). As our focus is on training on simulated data to produce strong performance on real data, we argue that a drop in synthetic performance is acceptable. Still, in case the reviewer is interested, we hypothesize that the introduction of augmentations that are (by design) out of distribution for the synthetic data causes this performance drop as the model is less able to overfit to specific synthetic-only features that increase performance on a synthetic test set.
> > >
> > > |  | Method | mIoU - GTAV Test (Source) |
> > > |---|---|---|
> > > | 1|Baseline |  73.45 |
> > > | 2|Baseline + RandAug | 68.78  |
> > > | 3|Baseline + PASTA | 70.27  |

---

> ### Comment · Reviewer_tiZd · 2022-11-21
> **Post-Rebuttal Comments:**
>
> Thanks for providing the detailed answers to my review comments. Based on your answers, I have some additional points to clarify:
>
> 1. Literature comparison: Regarding FSDR, I am not quite sure about the authors' argument that they are better because they use more data (which is also synthetic). In Table 3 of FSDR, the authors of FSDR described in the caption that they achieved 43.1 averaged mIoU on by training on GTA and testing on C/M/B, which is the same setting as Table 1-b) of this paper. PASTA is worse than FSDR, especially worse in GTV to Cityscapes. Also, I would like to add: Even they used both GTA and SYN together for training, it would still be a domain generalization setup for syn2real. So the authors could also compare with them in the same setup.
>
> 2. No strong performance gain in object detection: The authors argued they achieved strong gains over the baseline which is a rather vanilla of object detection training. If comparing Baseline + PASTA with Baseline + RandAug (which is a prior art on data augmentation), the gain vanishes with the RN-101 as the backbone. In this sense, I wonder if any randomization based augmentation method would be better than PASTA. Or, maybe more relevant, if the authors can show PASTA can provide complementary gain over RandAug, e.g., applying both of them.
>
> 3. Hyperparameter choices: My intuition is: the choice of hyperparameter can be quite sensitive to your target domain, i.e., C/B/M, and tasks (SemSeg/Object detection). Basically, it is up to your target domain high-frequency component distribution. For instance, if the authors can sweep through the hyperparameters, then respectively report mIoU on C/B/M, we can see if they share some common optimal choices of the hyperparameters. To me, as we dont have access to the target domain data, it would not be so nice the hyperparameters are sensitive to them.

---

> > ### Author Response · Authors · 2022-11-25
> > **Post-Rebuttal Clarifications**
> >
> > Thanks for responding to our rebuttal. We address individual points of clarification below.
> >
> > >Literature comparison: Regarding FSDR, I am not quite sure about the authors' argument that they are better because they use more data (which is also synthetic). In Table 3 of FSDR, the authors of FSDR described in the caption that they achieved 43.1 averaged mIoU on by training on GTA and testing on C/M/B, which is the same setting as Table 1-b) of this paper. PASTA is worse than FSDR, especially worse in GTV to Cityscapes. Also, I would like to add: Even they used both GTA and SYN together for training, it would still be a domain generalization setup for syn2real. So the authors could also compare with them in the same setup.
> >
> > Thanks for the suggestion. FSDR trains on the entirety of GTAV, amounting to ~12k extra training images compared to the experiments we ran for PASTA. As requested by the reviewer, in the table below, we compare Baseline + PASTA ($\alpha=3$) trained on the entirety of GTAV with FSDR (results from Table 3 in FSDR) and find that PASTA outperforms FSDR on all individual real target datasets. We will include these results in the final version.
> >
> > |  | Method | C |B | M | mean mIoU (Real) |
> > |---|---|---| ---| ---| ---|
> > | 1|FSDR | 44.80  | 41.20 | 43.40 |  43.13 |
> > | 2|Baseline + PASTA | **45.74** | **42.57** | **49.56** | **45.96**  |
> >
> > (GTAV$\rightarrow${Cityscapes, BDD-100K, Mapillary}; DeepLabv3+ RN-101)
> >
> > >No strong performance gain in object detection: The authors argued they achieved strong gains over the baseline which is a rather vanilla of object detection training. If comparing Baseline + PASTA with Baseline + RandAug (which is a prior art on data augmentation), the gain vanishes with the RN-101 as the backbone. In this sense, I wonder if any randomization based augmentation method would be better than PASTA. Or, maybe more relevant, if the authors can show PASTA can provide complementary gain over RandAug, e.g., applying both of them.
> >
> > Thanks for the suggestion. In the table below, we present syn-to-real generalization results for object detection (Sim10k$\rightarrow$Cityscapes) to assess if PASTA provides complementary gains over RandAug. We find that applying PASTA + RandAug provides significant gains over both PASTA and RandAug applied individually (across both RN-50 and RN-101). We will include these results in the final version.
> >
> > |  | Method | mAP@50 (RN-50) | mAP@50 (RN-101) |
> > |---|---|---| ---|
> > | 1|Baseline | 39.4  | 43.3 |
> > | 2| + RandAug | 52.8 | 57.2 |
> > | 3| + PASTA | 56.3 | 55.2 |
> > | 4| + RandAug + PASTA | **58.3** | **59.9** |
> >
> > >Hyperparameter choices: My intuition is: the choice of hyperparameter can be quite sensitive to your target domain, i.e., C/B/M, and tasks (SemSeg/Object detection). Basically, it is up to your target domain high-frequency component distribution. For instance, if the authors can sweep through the hyperparameters, then respectively report mIoU on C/B/M, we can see if they share some common optimal choices of the hyperparameters. To me, as we dont have access to the target domain data, it would not be so nice the hyperparameters are sensitive to them.
> >
> > We would like to clarify that in all our experiments, we change hyper-parameters only when the **source synthetic domain** changes -- i.e., from {GTAV, Sim10k} to VisDA-C (syn) -- and leave it fixed with a single model for all real target domains. As such, for our single-source to multi-target generalization settings, the choices of hyper-parameters are shared across all target domains. We restrict ourselves to this assumption because, in practice, it is unreasonable to assume access to the **right** labeled validation sets to select optimal hyper-parameters for every target domain.

---

> > > ### Comment · Reviewer_tiZd · 2022-11-30
> > > **Follow-up comment**
> > >
> > > Thanks for providing additional results.
> > >
> > > The new results to Q1 & 2 are quite positive, and interesting. Given that, I would suggest the authors to revise the experiment setup and comparison schemes. For instance, it is unclear to me why not using all available training samples from GTAV if you can. One advantage of synthetic data is: They can be generated and annotated with low cost. Also, for comparing with other data augmentation strategies, if combination provides good performance boost, it would be interesting to add discussion on potential reasons.
> > >
> > > Q3: yes, it should be a fixed single setup for all domains. BTW, how did you choose it ? based on the source domain validation performance? The intention of my previous question was to see if different target domains actually prefers similar hyperparameters, so an ablation study rather than a scheme for hyperparameter choice. Moreover, I am interested in if different target domains from real world have some similarity in terms of high frequency components, e.g., they have similar bandwidth. Overall, I think it is not difficult to get real world data, usually the bottleneck is more about annotation and curation effort. Therefore, if you can set and tune your frequency domain perturbation based on some spectrum analysis of real world data (without labeling), it may lead to more benefits.
> > >
> > > I will increase my score to 5. I believe this work has good potential to be accepted after proper revision.

---

> > > > ### Author Response · Authors · 2022-12-07
> > > > **Response to remaining comments and concerns**
> > > >
> > > > Thanks for the follow-up comments. We are pleased that the reviewer thinks our work is promising. We respond to follow up comments below.
> > > >
> > > > >The new results to Q1 & 2 are quite positive, and interesting. Given that, I would suggest the authors to revise the experiment setup and comparison schemes. For instance, it is unclear to me why not using all available training samples from GTAV if you can. One advantage of synthetic data is: They can be generated and annotated with low cost.
> > > >
> > > > We agree that synthetic data can be generated and annotated with low cost. Our reason for splitting source synthetic data from GTAV into reasonable training and validation splits is "model selection" -- after training for a fixed number of iterations, we select the checkpoint that achieves best in-domain (synthetic) validation performance. Relying on source domain validation performance for model selection is a common practice in domain generalization (and is consistent with prior work [A,B]), which in turn, almost always requires having access to a held-out split of source data. Since restricting ourselves to this reasonable assumption led to considerable improvements, we did not consider expanding beyond this. In the final version, we will include a discussion surrounding "more training data" and "model selection".
> > > >
> > > > [A] - RobustNet: Improving Domain Generalization in Urban-Scene Segmentation via Instance Selective Whitening
> > > >
> > > > [B] - In Search of Lost Domain Generalization
> > > >
> > > >
> > > > >Also, for comparing with other data augmentation strategies, if combination provides good performance boost, it would be interesting to add discussion on potential reasons.
> > > >
> > > > We believe RandAugment and PASTA produce different kinds of augmented views (also evident from Fig.1 in the main paper and Fig. 9 in appendix). The RandAugment vocabulary relies on a fixed vocabulary (and severity levels) of photometric operations but PASTA provides a structured way to perturb different frequency components. While the augmented views generated by RandAugment generally lead to improvements in several OOD settings, when operating with synthetic data that is deficient in high-frequency variations, PASTA augmented views are especially effective. Both of these augmentations, when put together, create views that are different from each of them applied individually. Consequentially, the RandAug + PASTA combination provides improvements over isolated applications of RandAug or PASTA. We will include a discussion surrounding the same in the final version.
> > > >
> > > > >Q3: yes, it should be a fixed single setup for all domains. BTW, how did you choose it ? based on the source domain validation performance? The intention of my previous question was to see if different target domains actually prefers similar hyperparameters, so an ablation study rather than a scheme for hyperparameter choice. Moreover, I am interested in if different target domains from real world have some similarity in terms of high frequency components, e.g., they have similar bandwidth.
> > > >
> > > > Yes. Out selection criteria are limited to operating with a **held out synthetic source domain validation split**. Different real target domains for our single-source to multi-target settings exhibit different high-frequency variation statistics (see Fig. 5 appendix) and therefore, it is possible for different real target domains to prefer different hyper-parameters.
> > > >
> > > > >Overall, I think it is not difficult to get real world data, usually the bottleneck is more about annotation and curation effort. Therefore, if you can set and tune your frequency domain perturbation based on some spectrum analysis of real world data (without labeling), it may lead to more benefits.
> > > >
> > > > We agree with the reviewer that while real data can be obtained, prohibitive constraints are usually associated with gathering and annotating data that is representative of real-world scenarios. However, we would like to humbly clarify to the reviewer that in this paper, we are interested in developing a method that provides strong out-of-the-box generalization on multiple real world scenarios when trained on synthetic data (domain generalization). Exploring whether additional labeled / unlabeled real-data (for training, hyper-parameter / model selection) can lead to more performance improvements or benefits is a domain adaptation problem (akin to FDA[C]) and is beyond the scope of this paper.
> > > >
> > > > [C] - FDA: Fourier Domain Adaptation for Semantic Segmentation

---

### Official Review · Reviewer_vBft · 2022-10-25

**Confidence:** 4
**Correctness:** 3
**Technical Novelty And Significance:** 2
**Empirical Novelty And Significance:** 2
**Recommendation:** 5

**Clarity, Quality, Novelty And Reproducibility:**

The methodology is clear and easy to reproduce with the content of the paper. However, while this method is considered novel, it is too naive to be a significant development.

**Strength And Weaknesses:**

** Strength
- This simple augmentation was effective in terms of accuracy on various tasks.
- The observation that high frequency signal variations in the synthetic images are smaller than in the real images is interesting.

** Weaknesses
- The proposed method is so naive to grab attentions of researchers in ICLR.
- There are various ways to increase the variation of the high frequency signal to take advantage of interesting observations. For example, increasing the signal size can be a method. Among these methods, what is the reason for using perturbation?

**Summary Of The Paper:**

This paper introduces the data augmentation method to better utilize synthetic images in model training for various tasks. Here, an augmented image is generated by perturbing the synthesized image, wherein the perturbed signal is selected from a normal distribution having a large standard deviation in proportion to a high frequency band. This perturbation is applied to the amplitude signals after transforming synthetic image to the frequency domain using FFT and retransformed after the perturbation. This method was developed based on the observation that the high-frequency signal variations in the synthesized image are relatively smaller than in the real image. Using these naive augmentation methods has been effective in significantly improving accuracy in various tasks such as semantic segmentation, object detection, and object recognition.

**Summary Of The Review:**

The proposed method has both the advantages of improving performance and the disadvantages of being developed too naively and not compared with other methods. I'm on the negative side now because I think the disadvantages outweigh the advantages, but my status may change after the discussion period with the authors is over.

---

> ### Author Response · Authors · 2022-11-12
> **Response to Reviewer vBft**
>
> We thank the reviewer for their feedback and respond to specific concerns below.
>
> #### **Simplicity of Method vs Contribution**
> >The proposed method is so naive to grab attentions of researchers in ICLR.
>
> >However, while this method is considered novel, it is too naive to be a significant development.
>
> We respectfully disagree with the notion that simplicity precludes novelty or significance. We believe one of the key strengths of PASTA lies in its simplicity and effectiveness in providing strong out-of-the-box generalization performance without any specialized components, task-specific design, or changes to learning rules.
>
> We would like to reiterate (see Sec 5.1 for further discussion) that across 5 syn-to-real shifts and 3 tasks, PASTA significantly outperforms a baseline model and consistently improves or is competitive with prior approaches that leverage additional advantages like more data or require specialized architectural components and objectives. Instead, we provide a simple augmentation strategy that is well-motivated from empirical observations and provides considerable improvements without changing anything about the model, architecture, or learning rules. We hope the simplicity of use (just modify your data augmentation pipeline) will enable fast adoption by the community and further exploration in this research direction.
>
>
> #### **PASTA Design**
> >There are various ways to increase the variation of the high-frequency signal to take advantage of interesting observations. For example, increasing the signal size can be a method. Among these methods, what is the reason for using perturbation?
>
> We would request the reviewer to elaborate on what they mean by "increasing the signal size". Our response is assuming the reviewer is referring to perturbing the raw RGB signal (please let us know if this was not the intent).
>
> We focus on increasing the variance of the Amplitude image as opposed to the raw RGB signal following a large body of literature that demonstrates that the image content is contained in the phase spectrum ([D-G] below; also outlined in Sec 2 of the main paper). Thus, to avoid inadvertently destroying semantic content necessary for solving a predictive task, we focus instead on perturbations in the amplitude spectrum, as prior work has explored. Our key contribution is in the design of a new method for applying random perturbations with structured variance (Eq 4/5) that can be applied to the amplitude spectra of synthetic data such that models trained on these augmented images perform well out-of-the-box on real data. Our design is carefully constructed following the observation that synthetic images have lower variance in the high-frequency components of the amplitude spectra compared to real images. For a more detailed discussion of our specific design, please see our comment to Reviewer nJC4.
>
> [D] - Phase in speech and pictures (ICASSP 1979)
>
> [E] - The importance of phase in signals (IEEE Proceedings 1981)
>
> [F] - A demonstration of the visual importance and flexibility of spatial-frequency amplitude and phase (Perception 1982)
>
> [G] - Structural sparseness and spatial phase alignment in natural scenes (JOSA A 2007).
>
> >The proposed method ...  the disadvantages of being developed too naively and not compared with other methods.
>
> Regarding comparisons with other frequency augmentation methods, please refer to our response to Reviewer tiZd.

---

### Official Review · Reviewer_nJc4 · 2022-10-27

**Confidence:** 3
**Correctness:** 3
**Technical Novelty And Significance:** 3
**Empirical Novelty And Significance:** 3
**Recommendation:** 8

**Clarity, Quality, Novelty And Reproducibility:**

I think that the current manuscript has no problems in terms of clarity, quality, and reproducibility.

**Strength And Weaknesses:**

[Strength]

S1. The problem dealt with in this paper is significant. Neural network models are getting larger and larger, and their training would require a huge amount of training data. Employing synthetic images as training examples is one of the promising solutions for this situation. Technically solid methods for filling the gap between synthetic and real images will draw attention from a broad range of researchers and engineers.

S2. The proposed method is simple but effective, which will be good news for practitioners interested in large-scale model training with a few training examples. Also, the proposed method works well for many applications, such as object recognition, object detection, and semantic segmentation.

S3. The documentation is clear and the proposed method is well justified.

[Weakness]

W1. I am not sure whether the variance given in Equation (5) is the best for this purpose. I understand that the proposed setting is better than the constant variance from the experimental results. However, I could find any theoretical justifications for this setting.

**Summary Of The Paper:**

This paper deals with the problem of synthetic-to-real domain generalization and proposes a method for augmenting synthetic images. The proposed method first derives power and phase components from a given source image and adds random jitters onto only power components for generating a pseudo-real image. A variance of random jitters increases as a spatial frequency increases, which indicates that high-frequency components are perturbed more than low-frequency components. Although the proposed method is simple, it outperforms baselines and several possible competitors in synthetic-to-real domain generalization.

**Summary Of The Review:**

I have a positive feeling about this paper since the proposed method is simple and effective and the documentation is clear. Also, as far as I  know, the proposed method for synthetic-to-real domain generalization is novel.

Other minor comments:

- The following paper can be regarded as an example of random amplitude perturbations. I do not think that this paper is critical for justifying the novelty of the proposed method.
"Adversarial Bone Length Attack on Action Recognition" (AAAI2022) https://ojs.aaai.org/index.php/AAAI/article/view/20132

- I am interested in the use of generative models for the same purpose.

---

> ### Author Response · Authors · 2022-11-12
> **Response to Reviewer nJc4**
>
> We thank the reviewer for their feedback and respond to specific concerns below.
>
> #### **PASTA Design**
> >I am not sure whether the variance given in Equation (5) is the best for this purpose. I understand that the proposed setting is better than the constant variance from the experimental results. However, I could find any theoretical justifications for this setting.
>
> Thanks for bringing this up. For ease of reference, we provide Equations (4) and (5) from our main paper which specifies our amplitude perturbation approach below.
>
> $g_{\Lambda}(\mathcal{A}(x))[m,n] = \epsilon[m,n] \mathcal{A}(x)[m,n]$ **(Eqn 4)**
>     where $\epsilon[m,n]\sim\mathcal{N}(1,\sigma[m,n])\mbox{ and }\sigma[m,n] = \left(2\alpha\sqrt{\frac{m^2 + n^2}{H^2 + W^2}}\right)^k + \beta \mbox{ and }\Lambda=\lbrace{\alpha, k, \beta\rbrace}$ **(Eqn 5)**
>
> We reiterate and attempt to clarify the goal of the three hyperparameters for this equation:
> * $\beta$: additive term, provides a constant level of variance. For semantic segmentation and object detection, we set $\beta$ to $0.25$. For object recognition, $\beta \in \lbrace{0.5,0.875\rbrace}$.
> * $\alpha$: a linear factor applied to the spatial frequencies. As frequencies get higher the variance increases linearly with $\alpha$. For semantic segmentation and object detection, we set $\alpha$ to $3$. For object recognition, $\alpha \in \lbrace{4,10\rbrace}$.
> * $k$: An exponential factor applied to the spatial frequencies. For a fixed $\alpha$, since the frequency term being exponentiated in Eqn 5 is normalized ($2\sqrt{\frac{m^2 + n^2}{H^2 + W^2}} \in [0, 1]$), increasing $k$ perturbs the lower frequency components less. $k$ is set to $2$ in our experiments.
>
> Our equation for computing amplitude variance that varies with the spatial frequencies was carefully designed to match the observation that **compared to real images, synthetic images have lower variance in the high-frequency components**. Thus, Eqn's 4 and 5 allow for augmented variance to be added to synthetic data that can expose a model to more high-frequency variations during training.
>
> *Note that setting either $\alpha =0$ or $k=0$ corresponds to removing any dependence of the variance on the frequencies.* Furthermore, for a vanilla baseline (not designed specifically for syn-to-real generalization) setting either $\alpha$ or $k$ very large ($\alpha > 4,k > 4$ for GTAV and VisDA-C with the RN-50 backbone) resulted in an expected performance reduction as the variance was too significant on the high frequencies in this case -- leading to severe augmentations that obfuscate task-relevant semantic information.
>
> As an illustrative example, consider the case of semantic segmentation from GTAV to {CityScapes, Mapillary, BDD-100K} (Real) using DeepLabv3+, RN-50, and 1024x560 resolution on GTAV. We can compare the performance of a baseline model without augmented images (row 1: 26.99) to our proposed PASTA approach (row 3: 41.90) which has a substantial improvement. If instead, we removed variance proportional to the spatial frequency (i.e., $\alpha=0$) we achieve a reduced performance (row 2: 39.19). However, it is important to note that the spatial variation should be proportional to spatial frequency (i.e., increase the variance as frequency increases). If instead, the variance is inversely proportional (more variance on the lower frequencies), this results in significant performance degradation (row 4: 8.9). These results are summarized in the table below.
>
> |  | Method | mean mIoU (Real)  |
> |---|---|---|
> | 1|Baseline  | 26.99  |
> | 2|+ PASTA ($\alpha = 0$) No spatial frequency dependence | 39.19  |
> | 3|+ PASTA ($\alpha > 0$)  Proportional to spatial frequencies | **41.90**  |
> | 4|+ Alternative Augmentation: Inversely Proportional to Spatial Frequencies | 8.90  |
>
> (Models trained at an input resolution of 1024x560)
>
> We would like to reiterate that our design of PASTA augmentation is motivated by observed phenomena in multiple synthetic and real datasets as well as empirically validated across multiple syn2real tasks, datasets, and deep architectures. Thus, we humbly argue our approach is of value and significance to the ICLR community.
>
> #### **Minor Comments**
> > Other minor comments
>
> Thanks for sharing the relevant pointer. We will discuss the same in related work. For future work, we agree that it might be beneficial to consider the extent to which perturbation schemes like PASTA can be extended to generative modeling/data. In particular, we think including PASTA-style perturbation schemes might be beneficial for diffusion-style generative models, since the underlying diffusion process involves stepwise perturbation of high / low-frequency components.

---

### Official Review · Reviewer_PoSx · 2022-10-28

**Confidence:** 3
**Correctness:** 3
**Technical Novelty And Significance:** 3
**Empirical Novelty And Significance:** 3
**Recommendation:** 6

**Clarity, Quality, Novelty And Reproducibility:**

Clarity and quality: The paper is written clearly.

Novelty: This paper provides some novel insights

Reproducibility: I believe this paper is simple to replicate.


**Strength And Weaknesses:**

Strength:
1. The proposed method appears to be reasonable. It is an intriguing idea to manipulate the amplitude spectrums of synthetic images in the Fourier domain for data augmentation.
2. The topic of this paper is interesting and important.
3. A large number of experiments are carried out to demonstrate the effectiveness of the proposed method.
4. The paper's writing is good.

Weaknesses:
1. Despite the fact that the authors conduct numerous experiments, they do not provide a sufficient discussion of the experimental results, so it is unclear how and why the proposed method works well. For example, for different tasks, the performance of the proposed method will change with different values of \alpha (\alpha is set to 3 in Tables 1-2 and to 10 in table 3). There is no information on how to set \alphal, or whether it can be directly set to 0 or 3 as mentioned in Section 5.2.
2. I think the proposed method does not be limited to synthetic data augmentation. It is preferable to discuss other application scopes, such as whether it can be used on generated data.


**Summary Of The Paper:**

This paper proposes a synthetic-to-real generalisation augmentation method called Proportional Amplitude Spectrum Training Augmentation (PASTA). The proposed method specifically augments synthetic data by perturbing the amplitude spectra, with magnitudes increasing with frequency. PASTA's effectiveness is demonstrated through experiments on three tasks and different settings.

**Summary Of The Review:**

Learning from synthetic data is an interesting and vital topic. This paper proposes perturbing the amplitude spectra of synthetic data for data argumentation, which may provide insights into how the community can better use synthetic data.

---

> ### Author Response · Authors · 2022-11-12
> **Response to Reviewer PoSx**
>
> We thank the reviewer for their feedback and address the specific concerns below.
>
> >Despite the fact that the authors conduct numerous experiments, they do not provide a sufficient discussion of the experimental results, so it is unclear how and why the proposed method works well. For example, for different tasks, the performance of the proposed method will change with different values of \alpha (\alpha is set to 3 in Tables 1-2 and to 10 in table 3). There is no information on how to set \alphal, or whether it can be directly set to 0 or 3 as mentioned in Section 5.2.
>
> * **Why PASTA helps**: Our motivating observation is that synthetic images are less-diverse in their high-frequency components compared to real images. Based on this, we specifically design PASTA to ensure more exposure to high-frequency variations when training on synthetic images. In Section A.4 of the appendix, we show how this discrepancy exists in multiple syn-to-real shifts across fine-grained frequency band discretizations of the amplitude spectrum (Figures 5 and 6). We further show PASTA helps counter this discrepancy (Figures 7 and 8) -- e.g., applying PASTA increases the standard deviation of the low, medium, and high-frequency components from 0.4 $\rightarrow$ 0.497, 0.33 $\rightarrow$ 0.51 and 0.3 $\rightarrow$ 0.52 respectively (more improvements concentrated on higher bands).
>
> * **Different optimal $\alpha, k,\beta$ across synthetic sources**: Synthetic images from GTAV and VisDA-C differ significantly - GTAV includes street view images whereas VisDA-C images are synthetic 2D renderings of 3D objects under different viewpoints and lighting [A] (Figure 9 in appendix). Compared to GTAV, VisDA-C images are relatively less diverse in high-frequency components. These differences contribute to the choice of optimal hyper-parameters.
>
> * **Overall sensitivity to $\alpha,k$**: Careful selection of $\alpha,k$ is more important compared to $\beta$. For our experiments, we find that restricting the $k\in[1,4]$ offers stable improvements. In addition to the observations in Section 5.2 for VisDA-C, we show the same in the following table for semantic segmentation GTAV$\rightarrow$Real, for DeepLabv3+ model (RN-50) backbone trained at a resolution of 1024x560. Setting $\alpha>0 \text{ } (\alpha = 3)$ aligns closely with our motivating observation and is always better compared to $\alpha = 0$ for all the settings we conduct experiments in. For $k\in[1,4]$, to obtain stable improvements for a vanilla baseline (not designed specifically for syn-to-real generalization), restricting $\alpha\in(0,4]$ offers stable generalization improvements. Exceeding these ranges leads to severe augmentations that obfuscate task-relevant semantic information. We include this discussion in appendix A.3 (in red).
>
> | | Method |mean mIoU (Real) |
> |---|---|---|
> |1 | Baseline|	26.99 |
> | | **Ablating $k$ for $(\alpha=3)$**|	 |
> |2 | + PASTA ($k=1$)|	41.50 |
> |3 | + PASTA ($k=2$)|	41.90 |
> |4 | + PASTA ($k=4$)|	41.09 |
> |5 | + PASTA ($k=6$)|	30.90 |
>
> (Models trained at an input resolution of 1024x560)
>
> [A] - Classification Track – http://ai.bu.edu/visda-2017/
>
> >I think the proposed method does not be limited to synthetic data augmentation. It is preferable to discuss other application scopes, such as whether it can be used on generated data.
>
> Great suggestion! For future work, we agree that it might be beneficial to consider the extent to which perturbation schemes like PASTA can be extended to generative modeling / data. In particular, we think including PASTA style perturbation schemes might be beneficial for diffusion-style generative models, since the underlying diffusion process involves stepwise perturbation of high / low frequency components.

---

### Author Response · Authors · 2022-11-12
**Thanks for the thoughtful feedback!**

We thank the reviewers for their thoughtful feedback and suggestions! We are pleased that they found (1) the paper to be well-written (R-PoSx) and clear (R-nJc4), (2) the problem setting interesting, important (R-PoSx), and significant (R-nJc4), (3) our motivating observation intriguing (R-PoSx) and interesting (R-vBft), (4) our proposed method simple and effective (R-PoSx, R-nJc4, R-vBft, R-tiZd), well-justified (R-nJc4) and (5) our experiments extensive (R-PoSx, R-nJc4).

We would like to re-emphasize that the key strength of PASTA lies in its simplicity (just modify your augmentation pipeline) and effectiveness in providing strong out-of-the-box generalization performance without any specialized components, task-specific design, or changes to learning rules -- as demonstrated by our generalization results across 5 syn-to-real shifts and 3 tasks.

We respond to individual reviewer comments in specific responses.

---

### Decision · Program_Chairs · 2023-01-20

**Decision:**

Reject

**Justification For Why Not Higher Score:**

Not sufficient support from the reviewers.

**Justification For Why Not Lower Score:**

NA

**Metareview: Summary, Strengths And Weaknesses:**

This paper proposes a data augmentation strategy, called PASTA,  for synthetic-to-real generalization, i.e., for improving the performance on real data of a model trained on synthetic data. PASTA involves perturbing the amplitude spectrums of the synthetic images in the Fourier domain to generate augmented views. It is motivated by the observation that the amplitude spectra are less diverse in synthetic than real data, especially for high-frequency components. Thus, PASTA augments synthetic data by perturbing the amplitude spectra, with magnitudes increasing for higher frequencies (this is where the novelty of the work lies).

The reviewers find the problem addressed interesting, the aforementioned observation insightful, the proposed method simple and yet effective, and the paper well-written.  However, there is a concern about the fact that PASTA is sensitive to hyperparameters and their choice is not discussed.  One possible way is to use a validation (real) domain, but this has not been tested.  Actually, it might not even be realistic to assume the availability of a separate validation domain in synthetic-to-real generalization. Other concerns include gains being not pronounced and even underperformed other data augmentation methods and the lack of comparisons with other data augmentation schemes.

While one reviewer votes for accept and another rates it marginally above the acceptance threshold, two other reviewers consider the work marginally below the acceptance threshold.


**Summary Of Ac-Reviewer Meeting:**

No one responded to a call for reviewer meeting.  The most positive reviewer wrote: "I do not want to fight against the reviewers and AC, which implies that we do not have an in-person meeting."